# The Role of the Serotonergic System in Time Perception: A Systematic Review

**DOI:** 10.3390/ijms252413305

**Published:** 2024-12-11

**Authors:** Rauf Akhmirov, Dina Mitiureva, Maria Zaichenko, Kirill Smirnov, Olga Sysoeva

**Affiliations:** 1Institute of Higher Nervous Activity and Neurophysiology, Russian Academy of Sciences, 117465 Moscow, Russia; akhmir5000@gmail.com (R.A.); dina.mitiureva@gmail.com (D.M.); mariya-zajchenko@yandex.ru (M.Z.); smirnov.kirill.sc@gmail.com (K.S.); 2Center for Cognitive Sciences, Sirius University of Science and Technology, 354340 Sirius, Russia

**Keywords:** time perception, interval timing, timing schedules, impulsivity, serotonin, 5-HT, serotonergic regulation

## Abstract

Time perception is a fundamental cognitive function essential for adaptive behavior and shared across species. The neural mechanisms underlying time perception, particularly its neuromodulation, remain debated. In this review, we examined the role of the serotonergic system in time perception (at the scale of seconds and minutes), building a translational bridge between human and non-human animal studies. The literature search was conducted according to the PRISMA statement in PubMed, APA PsycINFO, and APA PsycARTICLES. Sixty papers were selected for full-text review, encompassing both human (n = 10) and animal studies (n = 50). Summarizing the reviewed literature, we revealed consistent evidence for the role of serotonin in timing behavior, highlighting its complex involvement across retrospective, immediate, and prospective timing paradigms. Increased serotonergic activation appears to accelerate internal time speed, which we interpret through the dual klepsydra model as accelerated discharge of the temporal accumulator. However, some findings challenge this framework. Additionally, we link impulsivity—associated with decreased serotonergic functioning in our review—to a slower internal time speed. Variability in prospective timing tasks underscores the need for further research into how serotonin modulates reward-based temporal decisions, using novel approaches to disentangle internal time speed, response inhibition, and other factors.

## 1. Introduction

Time perception is a fundamental cognitive mechanism shared across species and crucial for the adaptive organization of behavior. The functional significance of the ability to perceive time is interconnected with anticipation of future events and realization of motor-intentional acts. Anticipatory pre-tuning of sensory and motor systems in order to implement actions allows living beings to achieve a positive adaptive result [1]. Theoretically, time perception can be seen as a proto-cognitive function, which is a prerequisite for other cognitive processes [2,3].

Time perception is a multifactorial process because there is no specific sensory system responsible for the time domain information. Genuine time perception is considered to be on the scale of several seconds to minutes because, on the millisecond scale, the representation of time domain information is determined by the properties of a particular sensory system involved. On a scale longer than several minutes, the information is determined by episodic memory [4,5,6]. Moreover, the scale of supra-seconds to minutes is believed to be the time scale of the subjective present [7].

Time perception is closely related to the concept of timing, which is widely used in experiments with animals. Timing (or interval timing for a duration of several seconds to minutes) refers to the temporal organization of behavior. At the same time, the term time perception refers to the aspect of the subjective experience of the passage of time and its duration [8], and this aspect can be inferred only indirectly and probabilistically from animal studies. In the current review, we will focus on the mechanisms underlying interval timing and time perception as the perception of duration on a scale of several seconds.

As in the domain of cognitive functions, in the domain of the brain and neuromodulatory processes, the mechanisms of time perception vary depending on the time scale [9,10,11,12]. Most studies on the role of dopamine in time perception point to its influence on intervals of less than 2 s [13]. At the same time, there is evidence to suggest the involvement of the serotonergic system in the time perception mechanisms on the scale of several seconds. This has been shown in pharmacological [14] and neurophysiological [15,16] studies of timing behavior in animals, as well as in pharmacological and genetic studies in humans (e.g., [17]). Nonetheless, it is important to note that both neuromodulatory systems are likely to be involved in sub- and supra-second timing to some extent [18], and the serotonergic system exhibits intricate relations with other neurotransmitter systems [9,15,19,20].

A range of comprehensive reviews on serotonergic modulation of time perception have been published to date. The review provided by the Bradshaw and Szabadi group highlighted the role of serotonin in animal timing behavior [14]. Subsequently, timing behavior has been studied in its relation to motivation, and how it is altered in diverse neuropathologies [21]. Marinho and colleagues explored genetic bases of time perception both in humans and animals [20]. Agostino and colleagues considered specific neural circuits with proposed neurotransmitter regulation [9]. Additionally, the effects of psychedelic treatments on timing have been discussed [22,23,24], offering a unique avenue for studying behavioral alterations through interventions with specific substances. However, the evidence towards serotonergic system involvement in time perception has not been systematically reviewed within the translational approach.

The translational approach is needed to comprehensively analyze existing findings on the role of the serotonergic system in time perception because there is a gap between interval timing studies in animals and time perception studies in humans. This gap exists mainly due to the extensive repertoire of invasive interventions in animal studies, which are highly limited in human studies, and theoretical distinctions in experimental paradigms. In the present review, we attempt to analyze evidence obtained in a wide range of timing schedules and paradigms and bind together the facts revealed in human and animal studies. It is of particular interest to examine whether the principles governing timing behavior in animals hold relevance in explaining time perception mechanisms in humans. Moreover, considering a significant number of studies conducted with novel and advanced methods of neural activity tracking, as well as neurochemical and molecular analysis, it is important to update the general overview of the topic and figure out the main gaps and unexplored areas for future research.

Therefore, the aim of the present systematic review is to assess the involvement of the serotonergic system in time perception according to the studies published to date. First, we will review existing evidence obtained from human studies. Second, we will review animal studies chronologically, grouping them depending on the experimental paradigm used. Then, we will discuss all the existing evidence on the role of the serotonergic system in time perception in a translational way and propose a concept of the role of the serotonergic system in time perception.

## 2. Materials and Methods

### 2.1. Literature Search Strategy

The literature search was conducted according to the PRISMA statement [25] in electronic databases such as PubMed, APA PsycINFO, and APA PsycARTICLES. PubMed was chosen for its extensive coverage of the biomedical literature, including neuroscience and mental health topics, which are central to our research focus. APA PsycINFO and APA PsycARTICLES were selected for their specialization in psychology and behavioral sciences, ensuring the inclusion of high-quality, peer-reviewed articles pertinent to our review. Together, these databases provide a robust foundation for identifying relevant studies across disciplines central to our research question. The database search was conducted in May 2023 (and refreshed in December 2024) with no restrictions as to the year of publication. The search terms were combined with Boolean operators as follows: ((“time perception”) OR (“interval timing”) OR (“sense of time”) OR (anticipation)) AND ((serotonin) OR (5-HT) OR (5-Hydroxytryptamine) OR (serotonergic)). Additionally, further papers (n = 21) were identified in the reference lists of the retrieved articles and screened for eligibility (backward snowballing). The search results from different databases were aggregated in the Nested Knowledge software platform. In total, after the removal of duplicates, 184 studies were found and proceeded to the screening procedure.

### 2.2. Eligibility Criteria

The screening procedure was conducted via the Nested Knowledge software platform and included a dual screening procedure by two assessors. Discrepancies among authors were settled through recurrent reviewing until a consensus was reached. Initially, the articles were screened for general eligibility according to their abstracts. Additionally, some articles were further excluded due to details revealed in the full text (see Section 3.1 below). A study was considered eligible if it was original experimental research that aimed to investigate the impact of 5-HT regulation on timing behavior, aspects of anticipation, or impulsive behavior as part of time perception. We did not include studies in which a timing scale irrelevant to the interval timing scale was investigated, such as millisecond, ultradian, or circadian scales.

### 2.3. Quality Assessment and Publication Bias

Quality assessment of both animal and human studies was conducted according to the Office of Health Assessment and Translation (OHAT) Risk of Bias Rating Tool for Human and Animal Studies [26]. The assessment was performed for each study to define a confidence rating with slight modifications of the items of the tool to better fit the studies (see details in the Appendix A). The overall value of the confidence rating is presented in the tables in the Results Section, and detailed tables with all quality assessment items are available in the Appendix A.

## 3. Results

### 3.1. Search Results and Study Selection

During the screening process, 75 papers were included in accordance with eligibility criteria and 109 were excluded during abstract screening (Figure 1). Among the excluded papers, 10 did not address serotonergic regulation in their experiments, 10 studies were devoted to irrelevant time scales (such as circadian rhythms), another 55 were not relevant because of the absence of timing or time perception aspects, 18 had non-timed anticipation (anticipation of something without a clear reference to a particular time scale), 14 were studies without empirical data (including 11 reviews or commentaries on scientific papers that did not contain experimental results), and the full text could not be found for the remaining two studies.

Additionally, 15 studies were excluded during the extraction of their full texts because careful reading of these articles elucidated the absence of related timing experience during experiments. Among them, animal studies used timing protocols or tests to evaluate anticipation, but further analysis of timing parameters was not included (n = 10). Similarly, among human studies, there were outcomes that were not directly relevant to the sense of time, such as in the monetary incentive delay task (n = 4, [27,28,29,30]) or four-choice serial reaction time task (n = 1, [31]). This paradigm is used to study the different stages of reward-based learning, from reward anticipation to its delivery, and the processing of reward and punishment, instead of parameters of timing or time perception, so studies with this experimental design were excluded.

Most studies included in the review involved experiments on rats (n = 42), with a minority involving other animals (six with mice, one with chicks, and one with rhesus monkeys). Results from these studies are described in the Animal Studies Section 3.4. The Human Studies Section 3.3 provides the main results extracted from papers implicating human participants (n = 10).

To provide a better structure for this review, we classify both human and animal studies according to the immediate relation of a subject to the interval to be assessed, as time intervals can be perceived differently depending on whether one anticipates waiting for a duration in the future, is currently navigating through it, or has recently experienced it. This classification is common for interval timing schedules and divides them into retrospective, immediate, and prospective [32,33].

Retrospective tasks refer to the recent past as a subject has to make judgments concerning a recently passed interval or intervals. This includes the interval bisection task (IBT) in animals and the duration discrimination task (DDT) and duration estimation (DE) in humans. In the translational context, DE is less applicable because it implies using conventional time units. At the same time, the human DDT and animal IBT allow some unhindered comparisons. In the IBT, animals are trained for interval categories, such as short and long, and then categorize the presented intervals, while in the DDT, participants are to compare consequently presented intervals. In humans, bisection tasks are also used (e.g., [34]), but they are rare and do not meet the criteria of inclusion for the current review.

The second is the immediate group of tasks in which a subject has to make a judgment concerning a currently passing interval. Such a scheme appears in the differential reinforcement of low rate (DRL), free-operant psychophysical procedure (FOPP), and peaked interval (PI) paradigms in non-human animals and in duration production (DP) and duration reproduction (DR) in humans. The specificity of DP is that it utilizes conventional time units similar to DE. DR is characterized by a more pronounced confounding of motor response timing than the DDT [35]. However, these paradigms in their essence might be comparable with animal paradigms, such as the PI paradigm.

The third group is the prospective tasks. In these tasks, a subject makes a decision concerning intervals which will pass in the future. These paradigms are most applicable to study aspects of impulsive behavior. One example of such a task is delay discounting (DD). Taken together, immediate and prospective timing tasks contain a pronounced aspect of anticipation.

Therefore, our research aimed to investigate how the serotonergic system modulates interval timing across retrospective, immediate, and prospective paradigms. Regarding changes in time perception across different tasks, our prior focus was to identify them in terms of response accuracy, considering the direction of alteration on the timeline, and response precision.

### 3.2. Ways of Assessing Serotonergic Functioning Implicated in This Review, Including Description of Pharmacological Agents and Their Properties

Most of the papers included in this analysis (n = 52) used pharmacological interventions to influence serotonergic functioning. A list of substances and their properties are presented in Table 1.

In a series of studies (n = 12), neurotoxin 5,7-dihydroxytryptophan (5,7-DHT) selective to serotonin neurons was applied with the aim of disrupting outgoing serotonergic projections from median and dorsal raphe nuclei. The raphe nuclei are an essential 5-HT afferent in the CNS with widespread projections throughout the brain. Additionally, the disruption of specific elements within the 5-HT system, including the knockout of receptors or key enzymes such as SERT, as well as metabolizing ferments (MAOA and TPH2), exhibit possible consequences and provide insights based on molecular mechanisms.

Other included studies (n = 24) used drugs targeting 5-HT receptors, in particular agonists and antagonists of 5-HT1A, 5-HT2A, 5-HT2B, 5-HT2C, and 5-HT3 receptors, which all play diverse roles in the central nervous system. Except for 5-HT3 receptors, all these subtypes belong to the metabotropic (G-protein-coupled) ‘super-family’ [36,37]. For each subtype, general information is provided as they potentially influence interval timing and have been examined in detail.

The 5-HT1A receptor subtype is found in both presynaptic and postsynaptic regions, mediating excitatory neurotransmission. Its activation reduces neuronal activity by opening potassium channels and closing calcium channels. High densities of 5-HT1A binding sites are found in limbic brain areas, notably the hippocampus, lateral septum, and cortical areas, while they are barely detectable in the basal ganglia and cerebellum [38]. 5-HT1A receptor antagonists induce anxiogenic responses, while other related ligands could exhibit anxiolytic and antidepressant properties, influencing circadian rhythms, exploratory behavior, aggression, impulsivity, sociality, and empathy levels [39].

The 5-HT2A subtype, located in postsynaptic regions, is widely distributed throughout the body, including in the brain, gut, and cardiovascular system. High levels of 5-HT2A binding sites are found in many forebrain regions, particularly in cortical areas, the caudate nucleus, the nucleus accumbens, the olfactory tubercle, and the hippocampus [40]. 5-HT2A activation mobilizes intracellular calcium through phospholipase C and phospholipase A signaling pathways [41]. These receptors are responsible for the regulation of behavior and are involved in memory processing and many other cognitive processes [42]. Abnormal 5-HT2A activity is associated with several psychiatric disorders, including depression, schizophrenia, and drug addiction [43].

5-HT2B receptors mediate excitatory neurotransmission, reducing the activation of phospholipase C. They act as autoreceptors on 5-HT neurons and interfere with the serotonin transporter (SERT) [44]. Activation of 5-HT2B receptors in the CNS inhibits serotonin and dopamine uptake, potentially influencing behavioral effects [45]. In the brain, this subtype distribution is restricted to a few brain regions, particularly within the cerebellum, lateral septum, dorsal hypothalamus, and medial amygdala [46]. Nevertheless, research on 5-HT2B receptors in relation to time perception remains limited.

5-HT2C receptors encompass both presynaptic and postsynaptic subtypes, which are expressed in several limbic structures, including the hippocampus (especially enriched in CA3), amygdala, anterior olfactory, and endopiriform nuclei, as well as within the cingulate and piriform cortex. Overactivity of 5-HT2C receptors may contribute to depression and anxiety, with abnormally high expression observed in suicide victims [47]. Activation of the 5-HT2C receptor increases phospholipase C activity [48]. Behavioral responses associated with 5-HT2C receptor activation include hypolocomotion, hypophagia, anxiety, penile erections, and hyperthermia. 5-HT2C receptors have been shown to modulate tonic and phasic inhibitory control on mesolimbic DA neuron activity.

5-HT3 receptors, which are ligand-gated ion channels, are located in postsynaptic regions, mediating excitatory neurotransmission. Activation of 5-HT3 receptors triggers rapid depolarization and increased cytosolic Ca2+ concentrations [49]. The highest densities of 5-HT3 receptors in the CNS are present in the dorsal vagal complex and spinal trigeminal nucleus of the lower brain stem, as well as some regions of the spinal cord. The cerebral cortex, hippocampal formation, amygdala, and medial nucleus of the habenula comprising the limbic region are also rich in 5-HT3 receptors [50]. At the behavioral level, 5-HT3 receptor activation leads to psychosis, anxiety, withdrawal effects from drug abuse, and eating disorders [51].

Additionally, 12 of the included studies also employed various anxiolytics, antipsychotics, and antidepressants, including selective serotonin reuptake inhibitors (SSRIs) and serotonin reuptake inhibitors (SRIs). Besides the acute post-injection effects of increasing serotonin levels in the synaptic cleft, the chronic effect of these medications was assessed in a few studies.

In 17 studies, psychedelics and CNS stimulants were used. Among the psychedelic treatments, substances such as the highly selective 5-HT2A agonists 5CN-NBOH, LSD, and psilocybin were used. In addition, the effects of CNS stimulants such as d-amphetamine, methamphetamine, cocaine, and mCPP were assessed in behavioral tasks, since these substances are non-selective inhibitors of monoamine transporters.

For a more nuanced understanding, some studies employed optogenetic techniques (n = 1) for the activation or inhibition of specific brain regions implicated in 5-HT regulation. Additionally, electrophysiological recordings of cellular or regional activity (n = 1) provided a strong foundation for studying the dynamic interplay between serotonin and temporal processing.

Assessing serotonergic functioning in humans is more challenging and only a few studies in our review used pharmacological substance administration in this regard (n = 6). Other studies (n = 2) applied a genetic analysis of interindividual differences in serotonin neurotransmission. Two studies used indirect inferences about the function of the serotonergic system via electroencephalography (EEG)—intensity dependence of auditory evoked potentials (iAEP)—as it was suggested as a neurophysiological marker of central serotonergic functioning [52,53,54]. The relevance of the serotonergic system to time perception can also be assessed by studying clinical samples, such as those from patients with clinical depression or schizophrenia, where serotonergic activity disturbances can be suggested. Two articles from our review used this approach.

### 3.3. Human Studies

There are only a small number of studies that examine the role of the serotonergic system in time perception in humans. A summary of the studies included after the screening procedure is presented in Table 2. The studies under review used classical time perception experimental paradigms with the explicit comparison, estimation, production, or reproduction of time intervals. In total, duration estimation tasks (DE) were used in two studies, five studies used duration reproduction tasks (DR), three used duration production tasks (DP), one study used sensory–motor synchronization with repeatedly presented tones, two studies applied a duration discrimination task (DDT) with a two-alternative forced choice design, and two studies employed a delay discounting task (DD). We will roughly divide the studies by the employed methods and experimental paradigms to review the main findings. If the reviewed manuscript contains more than one experimental paradigm, appropriate parts of the article are discussed in the corresponding sections devoted to the particular experimental paradigm.

#### 3.3.1. Retrospective Timing Paradigms: Duration Estimation and Discrimination

The DE and DDT can be grouped together as they both imply judgments upon already passed intervals, which is also called retrospective timing. In the DE task, participants have to indicate the duration of the interval presented to them using conventional time units. In the DDT, particularly in the form of interval comparison, participants are to compare the durations of intervals, which are presented pairwise. Among these tasks, the DDT possesses fewer limitations as there is no need to use conventional time units, and this feature also makes it comparable with paradigms used in animal studies.

**Table 1 ijms-25-13305-t001:** List of substances used in included studies.

Type	Compound Name	Target	Ki, nM *	Interval Bisection Task	Free-Operant Timing Schedule	Peak Interval Procedure	Delay Discounting	Other
Selective neurotoxin	**5,7-DHT** (5,7-Dihydroxytryptamine)	Serotonergic neurons		R [55,56,57,58]	R [55,59,60,61,62,63,64]	R [65]	-	R: [66,67] DRL
agonist	**8-OH-DPAT** (7-(Dipropylamino)-5,6,7,8-tetrahydronaphthalen-1-ol)	5-HT1A	1.14	R [62,68]	R [55,62,64]	R [69,70]	R [71]	-
agonist	**Buspirone**	5-HT1A	21.1	R [72]	-	-	-	-
agonist	**DOI** ((±)-1-(2,5-dimethoxy-4-iodophenyl)-2-aminopropane hydrochloride)	5-HT2A 5-HT2C 5-HT2B	0.68 2.38 20.03	R [73,74] M [75]	R [61,76,77,78]	R [69]	R [79,80]	-
agonist	**5CN-NBOH** (4-(2-((2-hydroxybenzyl)amino)ethyl)-2,5-dimethoxybenzonitrile)	5-HT2A 5-HT2B 5-HT2C	0.81 30 42	M [75]	-	-	-	-
agonist	**LSD** ((6aR,9R)-N,N-diethyl-7-(tritritiomethyl)-6,6a,8,9-tetrahydro-4H-indolo [4,3-fg]quinoline-9-carboxamide)	5-HT2A 5-HT1A 5-HT2C	0.73 2.7 10.21	-	-	-	-	H: [81] DR R: [82] DRL
agonist	**Psilocybin** (4-phosphoryloxy-N,N-dimethyltryptamine)	5-HT2A	13	R [83]	-	-	R [84]	H: [85,86] DR R: [82] DRL
agonist	**m-CPBG** (1-(3-Chlorophenylbiguanide)	5-HT3	-	R [87]	R [88]	-	-	-
agonist	**Ro60-0175** ((2S)-1-(6-chloro-5-fluoroindol-1-yl)propan-2-amine)	5-HT2C 5-HT2B	2.3 5.1	-	R [89]	-	M [90]	-
agonist	**Quipazine** (2-Piperazin-1-yl-quinoline)	5-HT3 5-HT2A	3.39 59	R [87]	R [88]	-	-	-
agonist SERT inhibitor	**mCPP** (3-Chlorophenylpiperazine)	5-HT2C 5-HT2A 5-HT1A SERT	3.4 32 44 202	-	R [89]	-	-	-
antagonist	**WAY-100635** (N-(2-(4-(2-methoxyphenyl)-1-piperazinyl)ethyl)-N-(2-pyridinyl)cyclohexanecarboxamide)	5-HT1A D4 5-HT2B	0.39 16.4 24	-	R [55,89]	R [69,70]	R [71,79]	M: [91] DRL R: [92] FCN
antagonist	**Isamoltane** (1-(propan-2-ylamino)-3-(2-pyrrol-1-ylphenoxy)propan-2-ol;hydrochloride)	5-HT1A 5-HT1B	21 112	-	R [89]	-	-	-
antagonist	**Ketanserin** (3-[2-[4-(4-fluorobenzoyl)piperidin-1-yl]ethyl]-1H-quinazoline-2,4-dione)	5-HT2A	8.0	R [73,87]	R [61,88]	R [69,73]	R [79,80]	-
antagonist	**MDL-100907** ((S)-2-(6-Chloro-5-fluoro-indol-1-yl)-1-methyl-ethylamine)	5-HT2A	0.1	R [73] M [75]	R [76,78,89]	-	-	-
antagonist	**SB-242084** (6-chloro-5-methyl-1-((2-(2-methylpyrid-3-yloxy)pyrid-5-yl)carbamoyl)indoline)	5-HT2C	0.16	M [75,93]	R [89]	-	R [79]	-
antagonist	**RS-102221** (8-[5-(2,4-Dimethoxy-5-(4-trifluoromethylphenylsulphonamido)phenyl-5-oxopentyl]-1,3,8-triazaspiro [4.5]decane-2,4-dione)	5-HT2C	8.5	-	R [78]	-	-	-
antagonist	**MDL-72222** (Bemesetron)	5-HT3	-	R [87]	R [88]	-	-	-
antagonist	**Granisetron**	5-HT3	1.41	-	-	-	-	H [94] DP
antagonist	**Mianserine**	5-HT2C 5-HT2A 5-HT2B	0.63 2.8 20	R [72]				
atypical antipsychotic antagonist	**Clozapine**	5-HT2A 5-HT2B 5-HT2C	5.35 8.37 9.44	-	-	R [95,96]	-	-
Non-selective monoaminergic drug	**Lisuride**	DA-, 5-HT-, and NE-receptors	-	-	-	-	R [79]	-
SSRI	**Fluoxetine**	SERT	5.98	-	-	R [97]	-	R [98,99] DRL M [91] DRL
SSRI	**Fluvoxamine**	SERT	2.5	-	-	R [100]	C [101]	
SSRI	**Sertraline**	SERT	3.05	-	-	-	-	R [99] DRL
SSRI	**Paroxetine**	SERT	0.58	-	-	-	-	R [99] DRL
SSRI	**Zimelidine**	SERT	67.6	-	-	R [102]	-	-
SRI	**Imipramine**	SERT NET	5	-	-	R [102]	-	-
SRI	**Clomipramine**	SERT NET	0.14 0.32 ^1^	R [72]	-	R [102]	-	-
SRI	**Desipramine**	NET SERT	0.63 17.6	R [72]	-	R [100]	-	M [91] DRL
CNS stimulant	**d-amphetamine**	NET DAT SERT	-	R [72,74]	R [60,76,77]	R [102]	-	-
CNS stimulant	**Methamphetamine**	NET DAT SERT	-	-	-	R [97]	-	R [103] DRL
CNS stimulant	**Cocaine**	NET DAT SERT	-	-	-	R [97]	-	-
5-HT precursor	**Tryptophan**	-	-	-	-	-	H [104,105]	-

^1^ Ki for Norclomipramine; * Ki values found in the PDSP open-access database. Notes: Ki—inhibitory constant, IBT—interval bisection task, DRL—differential reinforcement of low rate task; FCN—fixed consecutive number task; DP—duration production; DR—duration reproduction; R—treatment applied to rats, M—treatment applied to mice, H—treatment applied to humans, C—treatment applied to chicks (Gallus domesticus), SERT—serotonin transporter, NET—norepinephrine transporter, SSRIs—selective serotonin reuptake inhibitors, SRI—serotonin reuptake inhibitor.

Two studies employed DE tasks. Portnova and colleagues [106] showed the absence of significant relationships between duration estimations (5, 10, and 15 s) and variants of genes related to the efficiency of serotonergic neurotransmission, such as 5-HT2a T102C, 5-HTT LPR, and MAOA-VNTR. The second study conducted by Mavrogiorgou and colleagues [107] employed a DE task with a wider range of intervals (1, 2, 5, 13, and 32 s) and showed, albeit with unjustified post hoc analysis, that patients with schizophrenia overestimated time intervals compared to controls. The link between the serotonergic system and the choice to include a sample of schizophrenic patients was argued to be the result of the serotonergic hypothesis of schizophrenia pathogenesis, which implies an exaggerated efficiency of serotonergic neurotransmission in such patients [108,109,110].

Another two studies used the DDT. The first study explored relationships between psychophysical parameters derived from behavioral data and variants of polymorphisms related to serotonergic system functioning (5-HT2a T102C, 5-HTT LPR, and MAOA VNTR) [111]. While completing the task, participants were to compare pairs of visual stimuli with durations ranging from 3.2 to 6.4 s. The estimated psychophysical parameters involved the point of subjective equality (PSE) and the kappa parameter (K), which measures the loss rate of the neural accumulator for time in the dual klepsydra model (DKM) [112]. The higher the K and the lower the PSE in this paradigm, the more pronounced the effect of subjective time shrinking of the first interval in a pair compared to the second when they are objectively equal in duration. In turn, the more pronounced this effect is, the faster the internal time speed. The results showed that genetic variants related to more efficient serotonergic neurotransmission are associated with a more pronounced subjective time shrinking effect, i.e., a lower PSE and a higher K. Specifically, these genetic variants involved the SS allele of the 5-HTT LPR, which implies lower 5-HT reuptake, “low-expression” variants of MAOA VNTR, which imply slower 5-HT degradation, and TT variant of the 5-HT2a T102C gene associated with a higher density of receptors.

The second study using the same experimental paradigm was conducted on samples of patients with major depressive disorder and neurotypical controls [113]. The patients were in remission and had not taken medications for at least 2 months prior to the study. The choice of this clinical sample was also justified with the serotonergic theory of depression, which assumes a lower efficiency of serotonergic neurotransmission in those prone to such conditions. Coming back to the results of the study, the analysis of psychophysical data revealed significantly higher, and more objective, PSE values in the depression group compared to controls, which points to the slowing of internal time. At the same time, the intensity dependence of auditory evoked potentials (iAEP) was higher in patients than in controls, which might imply decreased serotonergic neurotransmission [52,53,54]. In general, the iAEP is another indirect physiological source of evidence regarding the efficiency of serotonergic neurotransmission. The result further strengthens the assumption that patients with major depressive disorder exhibit internal time deceleration due to deficient serotonergic neurotransmission.

Taken together, the evidence obtained from the studies that employed retrospective timing tasks suggests that a faster internal time speed is related to more efficient serotonergic neurotransmission.

#### 3.3.2. Immediate Timing Paradigms: Production and Reproduction Tasks

Production and reproduction tasks both imply making judgments upon an immediately ongoing interval. In the DP, participants are to produce the period defined by an experimenter in conventional time units, e.g., minutes or seconds. In the DR, a duration is presented by means of external stimuli, e.g., an auditory tone or visual dot, and a participant is to reproduce this duration by a key press. Another task that can be referred to this group is the sensory–motor synchronization task, although it is mainly used to control motor aspects of timing.

The DP task was used in three studies under review. Leigh and colleagues [94] examined the influence of the 5-HT3 antagonist, Granisetron, on timing performance in 12 healthy male participants. The authors implemented DP in the form of a subjective minute and found no statistically significant effect of Granisetron on performance [94]. The previously mentioned study by Portnova and colleagues [106] used this paradigm with intervals of 1, 2, 3, 4, 5 s, and 1 min. The results demonstrated that only MAOA polymorphisms appeared to be associated with subjective minute production: the polymorphisms causing less efficient degradation of serotonin (higher serotonin function) were related to longer produced minutes. In the study by Medvedeva and colleagues [113], the production of a minute was also used, and there were no differences in subjective minute production between the patients with major depressive disorder and controls, suggesting that deficiency in depression might not be related to the storage of conventional time units.

The DR was employed in five studies. The first two assessed the effect of psilocybin, a 5-HT2A agonist, on time perception. Wittman and colleagues [86] used duration reproduction with intervals of several seconds (1.5, 2, 2.5, 4, 4.5, and 5 s) and a sensory–motor synchronization task with repeatedly presenting tones (inter-stimulus intervals of 0.7, 1, 2, and 4 s). It was shown that participants under psilocybin tend to under-reproduce long intervals (2.5–5 s) and not to synchronize with beats with inter-trial intervals longer than 2 s (increased percent of missed synchronizations with tones). This effect was evident 90 min after psilocybin administration and more pronounced for the high-dose treatment. However, the results might be affected by working memory deficits and subjective conscious state alterations caused by psilocybin, which were also assessed in this study.

Subsequently, Wackermann et al. [85] reanalyzed the data of this study and conducted an additional experiment with the same DR task and very low doses of psilocybin, which should not cause alterations in the conscious state. The analysis was held in line with the DKM and included calculations of the K. The higher the K, the more pronounced the subjective time shrinking, and behaviorally, this effect is observed as a tendency to under-reproduce intervals. A reanalysis of this study conducted by Wittman and colleagues [86] showed that both medium and high doses significantly increased the K at the time of the maximal psilocybin effect (90 min after administration). The effect of a high dose on the K was the highest and increased the K by 44%. The results of the additional experiment demonstrated a significant effect of a very low dose of psilocybin on the K, increasing the parameter by 88%. This dose did not cause significant perturbations in subjective conscious states; however, working memory performance, which was affected by psilocybin in a previous study, was not assessed in this study. In addition, a very low dose causing a more pronounced effect than bigger doses may seem illogical. The authors argue that this discrepancy might be caused by individual differences in sensitivity to psilocybin as two samples were not matched, e.g., by age. Specifically, the participants of the second study were much older, and thus they might be more susceptible to psilocybin [114,115].

Yanakieva and colleagues [81] examined the effect of a microdose of LSD using a similar DR paradigm with visual presentation instead of auditory. While the mechanism with which LSD affects the serotonergic system is slightly different, both psilocybin and LSD are known to act as agonists of 5-HTR 2A receptors. Considering this rough similarity in the effects of these treatments, the evidence obtained seems contradictory. Specifically, it was demonstrated that LSD causes over-reproduction of supra-second intervals (between 2 and 4 s). Nevertheless, it is important to mention that the study sample included elderly participants, who could possess neurochemical alterations related to age. Although the authors claim that the dose had no clear effect on subjective conscious states, there was a dose-dependent effect on participants’ subjective estimation of the “drug effect”, which increased gradually from baseline to 120–180 min after administration. Moreover, the exact time after which the DR task was conducted following LSD administration was not unified across participants and was around 170 min after administration on average. The authors noted that in later stages (just in these times after administration), LSD can act as an agonist of D2 dopaminergic receptors [116], and this could act as a potential confounder. Additionally, the parallel study design with between-group comparisons might not be optimal for investigating drug effects on time perception due to high interindividual variability.

The previously mentioned study by Portnova and colleagues [106] also implemented DR with intervals of several seconds (1, 2, 3, 4, and 5 s) to assess potential relationships with serotonin-related genes. However, there were no significant results. Another previously mentioned study by Mavrogiorgou and colleagues [107] also reported null results regarding potential differences in DR between the clinical samples (schizophrenia and depression) or correlations between performances in duration reproduced and the iAEP.

Cumulatively, most of the studies that employed immediate timing paradigms and obtained statistically significant results suggest that enhancement of serotonergic neurotransmission leads to an increase in the rate of internal time speed.

#### 3.3.3. Prospective Timing Tasks

The study by Tanaka and colleagues [105] examined the role of serotonin in reward prediction at different time scales by altering serotonin levels through dietary tryptophan regulation. Participants’ serotonin levels were modified with three tryptophan conditions (depletion, control, and loading), and their brain activity was recorded via fMRI while they performed a task requiring choices between small, immediate rewards and larger, delayed rewards.

The results indicated that serotonin differentially affects short- and long-term reward prediction in distinct areas of the striatum: low serotonin levels heightened activity in the ventral striatum, associated with short-term reward prediction, while high serotonin levels increased activity in the dorsal striatum, which is linked to long-term reward prediction. This supports the idea that serotonin modulates the brain’s time scale for evaluating rewards by engaging different regions of the striatum depending on serotonin availability.

Interestingly, despite these neural differences, tryptophan manipulation had no significant effect on participants’ actual choice behavior. Metrics such as the choice ratio for delayed rewards and reaction times showed no differences across the tryptophan conditions. This suggests that while serotonin levels influenced neural processing related to reward prediction, these changes did not directly alter observable decision-making behavior in the context of the study.

However, in their next study, researchers found a behavioral change caused by tryptophan depletion [104]. They slightly modified the experimental paradigm, which included a change in motivation (from sugary drinks to money) and variations in delays. With these modifications applied, it was shown that tryptophan depletion increased the preference for smaller and sooner rewards over larger and later rewards. The discount factor (a measure of the devaluation of future rewards) was significantly lower under the low-serotonin condition compared to normal or high serotonin levels. At the same time, the high-serotonin condition (loading) did not cause any significant behavioral changes in comparison to the control condition. In addition, serotonergic influence on reward preference and discount factor was demonstrated to be independent of the learning rate, variability of choices, or fatigue induced by serotonergic manipulations.

Taken together, these works provide insights into how serotonin might underlie impulsivity and self-control at a neural level, affecting the brain’s evaluation of immediate versus future rewards, as well as at a behavioral level.

#### 3.3.4. Psychological Questionnaires

Another valuable source of evidence in human time perception studies is subjective reports obtained via specialized psychological questionnaires. For example, Jasper’s questionnaire (JQ) of time passage subjective speeds allows one to assess the direction (slowing or fastening) of subjective time flow. Using this questionnaire, Portnova and colleagues [106] demonstrated that MAOA variants causing less efficient degradation of serotonin (higher serotonin function) appeared to be associated with a slower subjective speed of time passage, and 5-HT2A variants associated with lower receptor density (lower serotonin function) were associated with an increased speed of subjective time passage. Together, this evidence aligns with that obtained in the experimental paradigms because with decreased internal time speed/temporal accumulator discharge, the objective time seems to be passing faster.

Questionnaires used for the assessment of subjective drug effects sometimes also contain scales evaluating disturbances in subjective time. For instance, in the study by Wittman and colleagues [86], it was shown that psilocybin causes disturbances in subjective time, according to the subjective time sense scale of the Altered States of Consciousness questionnaire.

Mavrogiorgou and colleagues [107] used a newly developed and unstandardized questionnaire to assess disturbances in subjective time. They reported a negative correlation between the iAEP, a neurophysiological marker of central serotonergic activity, and self-reported disturbances in sense of time, as well as symptom severity in the schizophrenia group, indicating increased serotonergic transmission for patients with more time disturbances and negative symptoms. However, the direction of these disturbances (whether subjective time was slowing or fastening) was not assessed and the result was significant but not corrected for multiple tests.

To sum up, the included studies suggest a link between enhanced serotonergic system functioning and disturbances in subjective time. Nonetheless, while subjective questionnaires are unique to human experience and can link experimental research with our real-life subjective feelings, there are many pitfalls in the proper interpretation of such subjective measures, as well as difficulties in correlating them with particular time scales.

#### 3.3.5. Summary

In conclusion, the majority of studies point to the involvement of the serotonin system in time perception, with more pronounced serotonergic neurotransmission associated with an accelerated internal time speed. There are some discrepancies that might have been caused by differences in samples, study designs, and experimental paradigms used. The most consistent results were obtained for paradigms that do not require conventional time unit representation and use ranges of several seconds. In general, there is a significant lack of studies on this topic, and the generalizability of the findings may be questioned due to low sample sizes. Another point to mention is that human studies were focused mainly on the accuracy of time perception and how it is influenced by serotonin, while the measures of precision were not considered or showed null results (e.g., in sensory–motor synchronization). Studies in animals pay more attention to this aspect of timing. Therefore, the issue of serotonin’s role in time perception can be further clarified via analysis of evidence obtained in animal studies.

**Table 2 ijms-25-13305-t002:** A summary of the included human studies. * The results of the quality assessment are presented in Appendix A.

Article	Sample	Time Perception (TP) Experimental Paradigm|Modality and Interval Range	Non-Temporal Psychological Measures	Serotonergic System Functionality: Manipulations or Analysis	Main Results	Quality Assessment: Total Score *
Leigh et al., 1991 [94]	12 healthy males (age 19–46 years, mean—32)	DP (1 min, 1 trial).		Granisetron (160 μg/kg)	No significant effect.	8
Wittmann et al., 2007 [86]	12 healthy volunteers (6 females, aged 26.8 ± 3.6 years)	DR (1.5, 2, 2.5, 4, 4.5, and 5 s, audio-presentation in random order, and 8 repetitions for a stimulus); Sensory–motor synchronization with repeatedly presenting tones (0.7, 1, 2, and 4 s inter-stimulus intervals). STS scale of the ASC.	Subjective drug effects (ASC and AMR); Working memory performance (SSP).	Psilocybin (115 and 250 μg/kg) + placebo; TP at 0, 90, and 240 min after administration.	Under-reproduction > 4 s with 250 μg/kg at 90 min after administration; Missed synchronization at 2 and 4 s; Depersonalization, derealization, and STS scales of the ASC; ↓ SSP performance at 90 min with 250 μg/kg; Corr. between STS and SSP; Corr. (trend) between STS and under-reproduction of 4 and 4.5 s intervals.	17
Portnova et al., 2007 [106]	89 healthy volunteers (only females, active or ex-synchronized swimmers, aged 8–67 years)	DP (1, 2, 3, 4, 5 s; and 1 min); DE (5, 10, and 15 s); DR (1, 2, 3, 4, and 5 s); JQ		Genotyping: 5-HT2a T102C, 5-HTT LPR, and MAOA VNTR.	MAOA polymorphisms with ↓ serotonin degradation: ↑ subjective minutes ↓ subjective time passage speed (JQ); 5-HT2a polymorphisms with ↓ receptor density: ↑ subjective time passage (JQ).	11
Tanaka et al., 2007 [105]	12 healthy volunteers (only males, aged 22–25)	DD: LL-rewards (~10 s and 3.2 mL of an isotonic drink) and SS-rewards (~5 s and 0.8 mL of a drink).		Tryptophan (depletion—0 g, control—2.3 g per 100 g amino acid mixture, and loading—10.3 g); Examination of the influence on brain activity (fMRI).	No behavioral changes; ↑ ventral striatum activity for SS-rewards ↑ dorsal striatum activity for LL-rewards.	16
Schweighofer et al., 2008 [104]	20 healthy volunteers (only males, age not defined)	DD: LL-rewards (~15–57 s and 20 yen) and SS-rewards (~1.5–7.5 s and 5 yen).		Tryptophan (depletion—0 g, control—2.3 g per 100 g amino acid mixture, and loading—10.3 g).	↑ SS-reward choices with ↓ serotonin (tryptophan 0 g); No effect of serotonin on learning rate or variability of choice.	16
Wackermann et al., 2008 [85]	Experiment 1: re-analysis of the Wittmann et al., 2007 data.	DR (1.5, 2, 2.5, 4, 4.5, and 5s, audio-presentation); Calculation of the K; STS scale of the ASC.	see Wittmann et al., 2007.	see Wittmann et al., 2007.	↑ K at 90 min after administration by 16% for 115 μg/kg and by 44% for 250 μg/kg.	see Wittmann et al., 2007.
	Experiment 2: 9 healthy volunteers (4 females, mean age 48.2 years).		Subjective drug effect assessment (the ASC and AMR scales).	Psilocybin (12 μg/kg) + placebo; TP at 90 m after administration.	↑ K compared to placebo by 88%; No alterations in subjective conscious state, only AMR “introversion” ↑.	17
Sysoeva et al., 2010 [111]	44 healthy volunteers (only males, aged 22 ± 4 years)	DDT (3.2–6.4 s, visual presentation); Calculation of the K.		Genotyping: 5-HT2a T102C, 5-HTTLPR, and MAOA VNTR.	Genetic polymorphisms related to ↑ 5-HT transmission (SS of 5-HTTLPR, TT of 5-HT2a T102C, and LA of MAOA VNTR):↓ PSE and ↑ K.	15
Yanakieva et al., 2019 [81]	48 healthy older adults (44% females, aged 63 ± 5.65).	DR (0.8, 1.2, 1.6, 2, 2.4, 2.8, 3.2, 3.6, and 4 s, visual presentation).	Subjective drug effects assessment.	LSD (5, 10, and 20 μg) + placebo; TP at 170 m on average.	↑ Over-reproduction of time intervals >1.6 s compared to placebo (higher for the 10 μg dose); The effect was independent of self-reported drug effects.	15
Mavrogiorgou et al., 2022 [107]	34—DPR (13 females; aged 32.4 ± 9.8 years); 31—SCZ (9 females; aged 35.1 ± 10.7 years); 33—Control (17 females; aged 32.8 ± 14.3 years).	DE (verbal report, intervals of 1, 2, 5, 13, and 32 s, visual presentation) DR (1, 2, 4, 14, and 27 s, visual presentation) TQ	Symptoms severity assessment: HDS, BDI, the Positive and Negative Syndrome Scale.	iAEP	Group effect on Duration estimation (*p* = 0.057, tendency); Overestimation of duration in SCZ vs. Control; ↑ TQ scores in DPR/SCZ vs. Control; Positive corr. between pathological symptoms severity and TQ scores; ↑ serotonin function (iAEP): ↑ disturbances of time feeling (TQ) in SCZ; No corr. between the iAEP and TP performance.	10
Medvedeva et al., 2023 [113]	31—DPR (25 females; aged 32.4 ± 9.8 years); 30—Control (21 females; aged 27.6 ± 9.3 years).	DDT (3.2–6.4 s, visual presentation); DP (1 min).	Symptoms severity assessment (BDI).	iAEP	↑ PSE values in the DPR group; ↑ iAEP in the DPR group; No difference between the groups in interval production.	13

Notes: TP—time perception; iAEP—intensity dependence of auditory evoked potentials; ↓—less, slow, decrease; ↑—more, fast, increase; corr.—correlation; AMR—the Adjective Mood Rating questionnaire; ASC—the Altered States of Consciousness questionnaire; SSP—the Spatial Span test; STS—subjective time sense scale of ASC; JQ—Jaspers questionnaire of time passage subjective speed; SS-reward—smaller-sooner reward; LL-reward—larger-later reward; fMRI—functional magnetic resonance imaging; PSE—point of subjective equality; K—the neural accumulator loss rate measure calculated using the psychophysical data according to the Dual Klepsydra Model of time interval representation (see details in the main text); DPR—depressed patients; SCZ—schizophrenic patients; Control—neurotypical participants; TQ—The “time questionnaire”, with higher scores indicating “the test subject has more problems overall with experiencing time” (newly developed for the study and unvalidated); BDI—Beck Depression Inventory; HDS—Hamilton Depression Scale. * The details of the quality assessment are provided in Appendix A.

### 3.4. Animal Studies

In contrast to human studies that are limited in approaches with which to track serotonergic functioning, animal studies have a larger arsenal of methods. However, the problem arises from the issue of time perception, as animals need to be trained to provide a particular response that might be interpreted as an index of their timing ability. The foundation of the paradigms aimed at studying aspects of timing behavior in animals can be traced back to Pavlov’s studies on the conditioned time reflex [117,118,119]. These early endeavors used temporal durations within the minute range as conditioning cues [120]. Building upon this groundwork, Skinner extended these investigations by introducing fixed-interval schedules of reinforcement [121], which led to the development of the fixed-interval peak procedure paradigm (Section 3.4.3). These milestones not only laid the groundwork for contemporary research on time perception in animals but also underscored the significance of timing mechanisms in understanding various facets of behavior, e.g., impulsive choices. Indeed, currently timing processes can be extracted from these paradigms, typically used to study impulsivity, such as differential reinforcement of low rate (DRL) and delay discounting (DD).

The paradigms in this review are organized based on the onset of timing schedules relative to the response, specifically focusing on whether an interval was provided before, during, or after the response. It begins with retrospective timing schedules, where reinforcement of the response depends on the duration of an interval that has already passed, such as the interval bisection task or temporal discrimination (Section 3.4.1). The next are immediate timing schedules, where a subject provides a response upon an ongoing interval based on their previous experience of reinforcement and current state (temporal differentiation: free-operant psychophysical procedure (FOPP, Section 3.4.2), fixed interval peak procedure (abbreviated as PI—peak interval) (Section 3.4.3), and differential reinforcement of low rate (DRL, Section 3.4.4)). We also review one prospective timing schedule, where the animal makes discriminative responses based on intervals that follow the responses (inter-temporal interval choice or delay discounting (DD), Section 3.4.5). Towards the end of this section, there are paradigms that do not directly measure interval timing but consider the serotonergic modulation of time perception, which could contribute to the experimental outcomes (Section 3.4.6). If the reviewed manuscript contains more than one experimental paradigm, appropriate parts of the article are discussed in the corresponding sections devoted to the particular experimental paradigm.

Throughout experiments, researchers have used slightly varied parameters and terminology to assess changes in time perception. Some describe premature and delayed responses to stimuli, while others refer to these phenomena as under- or overestimation of the presented duration, attributing these observations to changes in the speed of subjective time flow. Additionally, terms such as “internal clock” are used by some, but not universally adopted by all authors. To overcome subjectivity in interpretation in the Results Section, we will only refer to the experimental parameters used without referring to increased or decreased internal time flow. This interpretation will be conducted in the Discussion Section.

We also conducted a quality and risk of bias assessment of the studies included (Appendix A).

#### 3.4.1. Interval Bisection Task

The interval bisection task (IBT) is used to assess an ability to differentiate between two stimuli varying in duration. This task is also called the duration discrimination task, but as it has some important differences from the duration discrimination task often used in humans to compare stimuli of different durations presented in a pair, we will call it the interval bisection task to underscore the difference. In this task, animals are given the reward when they press the appropriate lever after the presentation of single stimuli of various durations. During the training period, animals learn to associate short stimuli durations with pressing lever A and long stimuli durations with pressing lever B, as a reinforcement available only after such choices. Then, animals are to discriminate stimuli of intermediate durations, and the count of lever presses for each duration is used to plot a psychometric curve that helps to infer the representation of internal duration.

The key parameters in the IBT include:

**Bisection point** (D50): The point of time when the response probabilities on levers A and B are equal (%B = 50). The bisection point characterizes an animal’s tendency to over- or underestimate a particular duration.

**Slope** (ε): The steepness of the psychometric function that relates the proportion of “long” responses on lever B to the stimulus duration. A steeper slope indicates more precise temporal discrimination.

**Difference Limen**: The smallest difference in duration between two stimuli that can be reliably detected—the just-noticeable difference. This can be calculated as the difference in duration between D50 and the time at which an animal responds with “long” levers in 75% of the trials (%B = 75). This parameter is highly correlated with the slope.

**Weber fraction** (WF): The ratio of the Limen to D50. This is a measure of sensitivity to changes in duration and as a consequence of the response precision.

**Omissions**: The number of trials without any response provided. This measure might indicate less involvement in the task or decreased ability to wait until the trial ends, potentially due to increased internal time speed.

Over the past three decades, a wide array of experiments employing diverse substances and exposure types have been conducted to investigate the modulation of timing behavior within this paradigm (Table 3).

One of the first studies to apply the IBT that is related to serotonergic functioning is the study of Morrissey and colleagues [56]. They aimed to assess the impact of the dorsal and medial raphe nuclei on the ability to distinguish intervals with durations between 2 and 8 s. Rats lesioned with 5,7-DHT demonstrated a leftward shift in psychometric function, with decreases in D50 values without significant alterations in the slope or WF. Additionally, Graham and colleagues [122] tested discrimination abilities using shorter intervals (200 ms and 800 ms), but no significant differences were found for any of the examined parameters. The authors no longer utilized such short intervals in subsequent research because these intervals are on the edge of the interval timing scale.

In 1995, Ho with co-authors investigated the impact of a “poke requirement” modification on timing behavior in rats with serotonergic pathway lesions [57]. The task was modified so that after the presentation of the stimuli, animals had to press a flap covering the food tray to access the levers. A poke into the food tray led to the insertion of the levers, and rats were able to make a choice. This modification was applied to avoid random lever responses after stimuli presentation. Rats with lesioned raphe nuclei also exhibited a decrease in D50 under the standard condition, but this effect was attenuated under the modified “poke requirement” condition. Consequently, the effects of serotonin depletion were less pronounced in the modified version, but still significant.

Further, using a standard IBT protocol, Ho and colleagues [100] examined the effects of acute and chronic desipramine and fluvoxamine treatment. Acute administration of either drug did not significantly alter D50, even at doses that significantly affected the operant response rate. Chronic treatment twice daily, 7 days a week, for 29 days with desipramine (8 mg/kg, i.p.) and fluvoxamine (8 mg/kg, i.p.) similarly did not induce changes in D50 or WF. These findings highlight that these antidepressants alone could not change timing differentiation in rats, although they induced changes in other aspects of behavior.

Al-Zahrani and colleagues [58] conducted a study employing a delayed choice modification of the IBT. Once the rats achieved over 90% accuracy in the IBT, a delay of either 8 s (phase I) or 12 s (phase II) was introduced between the end of stimulus presentation and the insertion of the levers. The delay modification aimed to assess the impact of short-term memory on the choice of the side of the lever press. This modification caused a rightward shift in D50 towards longer durations in both groups with either sham lesions or raphe nuclei destruction. Interestingly, this effect was significantly more pronounced in the lesioned group compared to the control group. The introduction of post-stimulus delays also led to an increase in the WF, which did not exhibit a significant difference between the two groups in the original version of the task. This result suggests the role of serotonin in the performance of delayed tasks including effects on both the precision and accuracy of interval timing.

Next, Bizot [72] investigated temporal discrimination in rats through the administration of various psychoactive drugs, including those influencing serotonergic activity, clomipramine and desipramine (SERT inhibitors), mianserin (5-HT2a antagonist), and buspirone (5-HT1A partial agonist), as well as others such as haloperidol, diazepam, and d-amphetamine. Rats were trained for more than 60 sessions to press lever A for a short duration (6–9 s—L5) and lever B for a long duration (15–19 s—L20). Then, in test sessions, durations of 12 s were introduced, and the authors evaluated the ratio of lever presses reinforcing short durations for stimuli presented at short (%L5—CS5), intermediate (%L5—CS12s), and long (%L5—CS20s) intervals. The %L5–CS12s ratio is analogous to the D50 metric, where an increase in D50 corresponds to an increase in the preference of choosing a short-duration lever, especially for the intermediate stimuli, and vice versa.

The SERT inhibitors desipramine and clomipramine led to an increased preference for the lever reinforcing short durations. Furthermore, such an alternation of temporal discrimination was linked to an increased ratio of omission trials, assessed by the count of omission trials and the ratio of correct responses, which might indicate less involvement in the task or a decreased ability to wait until the trial’s end, potentially due to an increased internal time speed. Similar effects were found for some other drugs tested, such as diazepam, atropine, scopolamine, and haloperidol. Conversely, d-amphetamine and nicotine reduced the percentage of choices for the “short-duration” lever without altering the omission rate. This study also found a decrease in the choices of the short-duration lever for the short-duration trials under mianserin, but this effect was present only in the intermediate drug dose of 2 mg/kg and did not extend into longer intervals.

The group of Bradshaw and Szabadi used the 5-HT1A receptor agonist 8-OH-DPAT, which increased WF but had no effects on D50 [62]. Further experiments conducted on rats by this group implemented the discrete trials psychophysical procedure (DTPP), which is similar to the IBT but involves longer and variable stimulus durations during the training procedure. In particular, intermediate durations were included, with lever A reinforcing durations from 2.5 to 22.5 s and lever B reinforcing durations from 27.5 to 47.5 s. Investigating the role of 5-HT receptors in the discrimination of durations, Body and colleagues [68] used the 5-HT1A receptor agonist 8-OH-DPAT in rats with either sham lesions or raphe nuclei lesions via intra-raphe injections of 5,7-DHT. The results showed no significant differences in performance between the control and lesioned groups. Although the introduction of 8-OH-DPAT had no influence on D50, it resulted in a dose-dependent increase in the WF in both groups, implying impaired timing precision regardless of the 5-HT system’s integrity. This effect of 8-OH-DPAT is likely mediated by postsynaptic 5-HT1A receptors, since the destruction of raphe nuclei with 5,7-DHT, which causes the depletion of presynaptic 5-HT1A receptors, did not lead to any differences in timing performance between the experimental groups.

The subsequent study by Asgari and colleagues [73,87] evaluated the role of 5-HT2 and 5-HT3 receptors in the IBT. The authors demonstrated that the 5-HT2A receptor agonists DOI and quipazine interfered with temporal discrimination, with a rightward shift and flattening of the psychometric function, accompanied by notable increases in the slope and WF. These effects of the agonists could be counteracted by the 5-HT2A receptor antagonists ketanserin and MDL-100907. At the same time, the 5-HT3 agonist m-CPBG did not demonstrate any significant changes in psychometric function, and the 5-HT3 antagonist MDL-72222 could not attenuate timing behavior alterations induced by 5-HT2A activation. Thus, while modulation of 5-HT2A receptors affects the accuracy and precision of interval timing, 5-HT3 receptors apparently are not involved in the regulation of timing behavior.

Further, Hampson and colleagues [74] investigated the impact of d-amphetamine and DOI on rats’ ability to discriminate both durations and light intensities. The results indicated that these drugs had similar effects on temporal discrimination, increasing the WF and D50. However, it is noteworthy that d-amphetamine decreased light intensity discrimination abilities by increasing WF, while DOI did not. Thus, d-amphetamine showed more general effects on discrimination ability, probably related to attention, while DOI is more related to duration discrimination, supporting the previous study by Asgari and colleagues [73].

Recently, Popik and colleagues [83] investigated the effects of rapid-acting antidepressants on temporal discrimination in the IBT, assessing the mechanism of their therapeutic properties. The authors aimed to contrast the effects on a 5-HT2A agonist, psilocybin, its metabolite psilocin, and the novel psilocybin derivative, norpsilocin, with DA agonists, (S)- and (R)-ketamine, which are suggested to have opposite effects on internal time flow. The study showed that none of the forms of psilocybin, acting as 5-HT agonists, demonstrated any significant alterations in timing performance, while (S)-ketamine (15 mg/kg i.p.), as a more psychoactive isomer of ketamine, shifted the psychometric curve rightward. However, it also led to a significant decrease in the accuracy of temporal discrimination, an increase in latencies of response, and an increase in the number of incorrect responses. Thus, the observed effect of this psychoactive DA agonist on timing might be a drawback of generally poor performance in this task.

Several studies have examined timing behavior in mice using the IBT. Avlar and colleagues [93] investigated the relationship between motivation and timing performance in mice overexpressing striatal dopamine D2 receptors (D2R-OE group) and control animals. Based on previous research, the authors assumed that deficits in motivation were present in the D2R-OE group and reported that this group also showed deficits in temporal discrimination, reflected in a decreased slope and timing precision. Furthermore, the authors examined the effect of a selective 5-HT2C receptor antagonist, SB-242084, known for increasing incentive motivation in these animals. The study showed improved temporal discrimination, enhancing timing precision and accuracy in both the control and transgenic mice after SB-242084 treatment. In particular, the blockade of 5-HT2C receptors led to a decrease in D50 and an increase in the slope of psychophysical function. While this study underscores the role of motivation in temporal cognition, suggesting that enhancing motivation could be a powerful strategy for improving timing accuracy, as well as relating the effects to the dopaminergic system, the direct impact of the serotonergic signaling pathway on timing behavior can also be suggested from the reported results.

The study by Halberstadt and colleagues [75] further explored the effects of 5-HT2A and 5-HT2C receptor ligands on temporal cognition by utilizing the DTPP task in C57BL/6J mice. Consistent with previously reviewed work [93], the 5-HT2C antagonist SB-242084 also reduced D50. At the same time, the 5-HT2A antagonist M-100907 increased D50, pointing to the opposite effect. Surprisingly, the 5-HT2 agonist DOI and the selective 5-HT2A agonist 25CN-NBOH both also led to increased D50, additionally also increasing WF, highlighting that relative preference for the “short-duration” lever was supplemented by disruptive precision on timing. A similar D50 increase induced by both agonists and antagonists of 5-HT2A receptors is rather hard to explain unless a U-shaped relationship is suggested between 5-HT2A activation with duration representation, a hypothesis that has to be further examined in future studies. At the same time, the effect of DOI was antagonized by M-100907 but was unaffected by SB-242084. Taken together, these results suggest that both 2A and 2C receptor types participate in the regulation of temporal processing, influencing the internal representation of durations.

In conclusion, during the past three decades, the IBT and its modifications have been established as a versatile and effective behavioral paradigm for investigating interval timing. Various experimental aspects such as reinforcement magnitude, stimulus intensity, and different durations from milliseconds up to a minute were evaluated by comprehensive studies using this paradigm. This extensive examination has elucidated the principal involvement of ascending 5-HT projections in temporal discrimination, implicating both 5-HT2A and 5-HT2C receptors in this process. Thus, destruction of the raphe nuclei, as well as a 5-HT2C blockade, decreased D50. On the other hand, 5-HT2A activation and SSRI treatment increased it.

#### 3.4.2. Free-Operant Psychophysical Procedure

In the next paradigm, the free-operant psychophysical procedure (FOPP), animals react within the ongoing trial, not in relation to the interval that has already ended as in the IBT and can obtain reinforcement for responding on one of the two continuously available levers depending on the timing of the response. Each trial lasts for a preset standard period (typically in a range of 50–80 s), which is divided into two equal intervals. During the first interval, pressing of lever A is reinforced, and pressing of lever B is reinforced during the second half of the trial. The response rate on these levers changes over the course of the trial, with the response rate increasing on lever B while the response rate decreases on lever A. Response rate changes during the trial are used to fit a psychometric curve. The parameters of the psychometric curve are further analyzed and reflect timing performance.

The metrics assessed in this paradigm are the indifference time point (T50), the slope of the psychometric curve (ε), the Weber Fraction (WF), the Rate of switching between levers and the Switching Time (S50), with the latter parameters providing control for impulsivity.

**Indifference Time Point** (T50): T50 represents the time at which an animal is equally likely to respond on either of the two levers, A and B. In other words, it is the point in the trial where the response rates on both levers are balanced, and the subject is “indifferent” between choosing one over the other (percentage of responses on lever B (%B) = 50). T50 is a key measure of an animal’s perception of time intervals in the FOPP.

**Slope** (ε): The slope of the logistic curve/psychometric function, denoted as ε, describes the smoothness at which the response rate shifts from lever A to lever B during the trial. The slope of the logistic curve provides insights into how quickly animals adjust their responses in relation to the changing reinforcement availability, which could reflect the precision of timing. A steeper slope indicates a more rapid shift in responses between levers, while a shallower slope suggests a slower transition.

**Weber Fraction** (WF): The WF is a psychophysical measure that quantifies an animal’s sensitivity to changes between intervals during the trial. It is calculated as the ratio of the limen (half the difference between the durations corresponding to %B = 25 and %B = 75) to the duration of the reference time interval (T50). In the context of the FOPP, a higher WF implies lower precision when switching between the two levers.

**Switching Rate**: This parameter is usually assessed as a number of switching between the two levers within the experiment. The higher switching rate might reflect the low inhibitory regulation in switching between behavioural states.

**Switching Time** (S50): S50 refers to the time of inflection point for cumulative probability of a lever switching and provide an additional parameter of timing accuracy

Bradshaw and Szabadi’s research group conducted various experiments, changing parameters and types of exposures to investigate the modulation of timing behavior in this paradigm (Table 4).

AI-Zahrani and colleagues [59] and Chiang and colleagues [63] focused on the effects of raphe nuclei lesions on timing behavior using the FOPP. Lesions were conducted through the targeted destruction of the central serotonergic pathways via an intra-raphe injection of 5,7-DHT. While the indifference time point (T50) and the slope of the psychophysical function remained unaffected by the lesions, a noticeable increase in the rate of switching between response alternatives was observed in the lesioned group. This led to the suggestion that central serotonergic pathways might be involved in inhibitory regulation, specifically in the context of switching between behavioral states.

Chiang and colleagues [62] studied 5-HT1A receptor modulation of timing behavior. They employed a selective 5-HT1A agonist 8-OH-DPAT in two versions of the experiment. The “constrained” version introduced a novel protocol where the first press on lever B resulted in the withdrawal of lever A until the start of the next trial. In contrast, the second “unconstrained” version was akin to the protocol from the earlier study [59]. In the “constrained” version, significant reductions in T50 were observed at higher doses (100 and 200 μg/kg), while in the “unconstrained” version, a noticeable reduction of the T50 was observed across a wider range of doses (50, 100, and 200 μg/kg), which was also accompanied by an increase in switching rate at the maximal dose. Thus, by showing a similar effect for the constrained and unconstrained conditions, the authors demonstrated that the T50 shift is independent of the increase in the switching rate, eliminating its link to pure impulsivity.

Building upon this, Body and colleagues [55,64] expanded their exploration of 5-HT1A receptor modulation of timing behavior using agonists and antagonists of the 1A receptors with intact rats and those with disrupted serotonergic pathways due to 5,7-DHT injections. 8-OH-DPAT reduced T50 without significantly affecting the WF in both control and lesioned groups. The authors argue that the effects of this agonist could be mediated by postsynaptic 5-HT1A receptors, since presynaptic receptors were disrupted with 5,7-DHT. Notably, the application of the 5-HT1A receptor antagonist WAY-100635 blocked the effects of 8-OH-DPAT. Thus, these studies provide consistent evidence of the involvement of postsynaptic 5-HT1A receptors in the accuracy but not precision of interval timing in the FOPP, contrary to what was found in the IBT task reviewed above.

Continuing their research on receptor regulation, Body and colleagues [61] examined the effects of the 5-HT2 receptor agonist DOI and the 5-HT2A antagonist ketanserin on timing performance. In addition to T50 indifference point, they introduced the S50 parameter measured as the inflection time point in the cumulative switching rate function constructed over the trial duration. DOI dose-dependently reduced T50 and S50, and these effects were blocked by ketanserin. This led to the presumption that 5-HT2A receptors, rather than 5-HT2C, likely mediated the impact of DOI on temporal processing. It was verified in a subsequent study, in which Body and colleagues [78] compared the effects of agonists and selective antagonists for 5-HT2A and 5-HT2C receptors. Again, subcutaneous administration of DOI led to a significant reduction in T50 and a reduced response rate, which were effectively reversed by the administration of a highly selective 5-HT2A receptor antagonist, MDL-100907. However, intracerebral injections of DOI within the dorsal striatum, a region involved in the regulation of interval timing, did not produce significant effects on timing parameters. Furthermore, intracerebral injections of neither MDL-100907 nor the 5-HT2C receptor antagonist RS-102221 reversed the effect of subcutaneous DOI administration on performance in this task. These results suggest that the effects of DOI on timing do not appear to be mediated by both 5-HT2A and 5-HT2C receptors in the dorsal striatum and should work through other brain structures.

Another 5-HT2A receptor agonist, quipazine, also decreased T50 and S50 [88], akin to DOI treatment. This drug also has an affinity for 5-HT3 receptors, but evaluation of the ligand activity of the 5-HT3 receptor revealed that this type of receptor does not take part in interval timing. Treatment with the 5-HT3 agonist m-CPBG had no significant effects on T50, and administration of the 5-HT3 antagonist MDL-72222 did not attenuate the effects of quipazine.

The interplay between the involvement of serotonergic and dopaminergic pathways during treatment with agents with multiple effects was examined in the study of Body and colleagues [76]. The authors demonstrated that the effect of DOI was not regulated by DA receptors, since the D2 inverse agonist haloperidol and the D1 antagonist SKF-83566 did not attenuate the effects of the DOI treatment. At the same time, d-amphetamine administration reduced T50, but these effects were canceled by treatments with SKF-83566 or with the 5-HT2A antagonist MDL-100907. This study confirms that 5-HT2A and D1 receptors are involved in the shared mechanism of d-amphetamine’s effects in altering temporal differentiation.

Cheung and colleagues [60] shifted the focus towards exploring the effects of chronic treatment (up to 15 days) with DOI and its interaction with the non-selective psychostimulant d-amphetamine and the D2/D3 agonist quinpirole. This investigation extended the temporal procedures by including drug treatment conditions without concurrent behavioral training. The study demonstrated the development of tolerance to the effect of DOI on T50, but only when animals simultaneously underwent behavioral training, pointing out that this effect represents behavioral adaptation rather than pharmacological desensitization. Moreover, after DOI treatment, cross-tolerance to d-amphetamine was observed. At the same time, quinpirole decreased T50 in animals with tolerance to DOI, showing no cross-tolerance effects for D2/D3 activation, which implies that 5-HT2A and D2/D3 receptors have distinct regulation mechanisms regarding timing behavior.

A further study examined the sensitivity of interval timing in both intact and raphe nuclei-lesioned rats to various compounds such as d-amphetamine, quinpirole, and the D1 agonist SKF-81297 [60]. The D1 agonist SKF-81297 and the D2 agonist quinpirole reduced T50 in both groups, whereas d-amphetamine reduced it only in the control group. The results indicated that d-amphetamine’s effects on temporal differentiation required an intact serotonergic system, and the effects of DA selective agonists are provided by separate pathways. This evidence implies two different mechanisms underlying the effects of psychostimulants and selective ligands on timing performance.

Body and colleagues [89] studied the role of 5-HT2C receptors in interval differentiation in the FOPP. For this reason, a non-selective 5-HT receptor agonist mCPP, with a high affinity for 5-HT2C receptors, and the highly selective 2C agonist Ro-600175 were used. As a result, the more selective agonist Ro-600175 had no effect on T50, but mCPP consistently reduced T50. Notably, the reduction in T50 induced by mCPP was not antagonized by the selective 5-HT2C receptor antagonist SB-242084. Instead, the coadministration of mCPP and a 5-HT2C antagonist reduced T50 even more than mCPP alone. Given that mCPP has an affinity for various 5-HT receptor subtypes, including 5-HT1A, 5-HT1B, and 5-HT2A receptors, the study explored the role of these receptors in mediating mCPP’s effects. The results indicated that the 5-HT1A receptor antagonist WAY-100635 and the 5-HT2A receptor antagonist MDL-100907 partially reversed the effect of mCPP on T50, while the 5-HT1B receptor antagonist isamoltane was ineffective. This pattern of results suggests that the reduction in T50 induced by mCPP is likely mediated by the stimulation of 5-HT1A and 5-HT2A receptors. Interestingly, the role of 5-HT2C receptors in this context remains unclear, since 5-HT2C ligands do not influence T50 itself, but a 2C blockade enhances the effects of 1A and 2A activation on T50 reduction. Notably, the pharmacological effects on S50 was similar to those for T50, supporting the reported serotonergic influence on timing behaviour measured with different parameter.

In conclusion, these series of studies provide crucial insights into the nuanced effects of different ligands affecting serotonin functioning and the complex interplay between serotonin and dopamine systems in modulating timing behavior in the FOPP. The findings emphasize the roles of 5-HT1A and 5-HT2A receptors in the accuracy of interval timing in the FOPP. In the course of these works, consistent T50 decreases were found, mediated by the activation of 5-HT2A and 5-HT1A receptors and by d-amphetamine as a non-selective psychostimulant. The T50 decrease can be restored by the desensitization of 5-HT2A receptors. Additionally, the ligands targeting 5-HT2C receptors did not directly impact timing accuracy. However, the blockade of 5-HT2C receptors augmented the effects resulting from the activation of 5-HT1A and 5-HT2A receptors. Next, the reduced T50, caused by selective dopamine ligands, apparently has separate mechanisms of response in timing regulation as these parameters were not affected by the serotonin interventions. It should be noted that serotonergic agents showed opposite effects on T50 in the FOPP and D50 in the IBT. This effect might be due to the specifics of the tasks, as in the FOPP, animals are learning from ongoing behavior, with a need to constantly attend to the passage of time on which the reinforcement depends. In the IBT, animals have to rely on the interval that has already passed, which might affect the way interval representation is stored.

#### 3.4.3. Fixed Interval Peak Procedure (PI)

The fixed interval peak procedure, also known as the peak interval (PI) procedure, is a well-established experimental paradigm to investigate timing processes in animals. In this test, animals could obtain reinforcement after a preset time interval has elapsed. It is rooted in traditional Skinner’s fixed interval (FI) schedules, where the first response after a fixed duration terminates the stimuli and leads to a reinforcement [121]. In addition to this fixed-interval part, the PI procedure introduces “peak” trials without reinforcement or signals for an elapsed time. As a probe, the test includes up to 50% “peak” trials, which are randomly implemented in a session with reinforced FI trials. Behavior in these probe “peak” trials consists of progressively increasing the response rate up to the criterion interval, followed by a declining rate of response. This modification of FI schedules makes it possible to explore the whole distribution of animal response around critical intervals, providing more information on timing behavior. Thus, the PI procedure can be considered as a valuable tool for understanding an animal’s time perception and response strategy in time-based decision-making.

Usually, this test evaluates the peak response rate (PRR), T-peak, spread time, and WF as parameters for the assessment of timing behavior.

**Peak Response Rate** (PRR): The PRR represents the maximum rate of response observed during the PI procedure. As the interval progresses, animals gradually increase their response rate, reaching the maximum at the time when reinforcement is most likely to occur. The PRR serves as a measure of general activity level that might be related, e.g., to animal motivation.

**Time to Peak** (T-peak): The T-peak refers to the moment at which the response rate reaches its maximum within the PI interval. This metric measures the accuracy of an animal’s timing ability.

**Spread Time** (ST): The ST characterizes the width of the response distribution during the PI interval. It is conveniently measured as the time window within which the response rate reaches 70% of its maximum value. A narrower spread indicates a high level of temporal precision, whereas a broader spread suggests less precise temporal discrimination.

**Weber fraction** (WF): The WF is a parameter used to assess an animal’s sensitivity to temporal discrimination. It is calculated as the ratio of the ST to the T-peak. A lower WF indicates a higher sensitivity to temporal differences, signifying more precise temporal perception.

During the search, 11 articles were found for this paradigm, with nine conducted on rats and two on mice (Table 5).

Morrissey and colleagues [65] were the first to study the role of the dorsal and median raphe nuclei in timing behavior using this paradigm. The experiment was conducted on rats, half of which received lesions of the dorsal and median raphe nuclei, while the other half was a control group with sham lesions. The results revealed that the PRR and T-peak did not significantly differ between the two groups. Notably, the lesioned group exhibited a broader spread of the response rate function compared to the control group, and the ST/T-peak ratio related to the WF was significantly higher in the lesioned group. Thus, raphe nuclei destruction impaired timing precision, while the timing accuracy and inner clock speed remained intact in this experimental context.

In another study, the same group of authors examined the effects of acute and chronic treatment with selective SERT and NET inhibitors (fluvoxamine and desipramine) on the peak procedure [100]. In the acute treatment experiment, some of the rats underwent treatment with desipramine, and the rest underwent treatment with fluvoxamine. The PRR was significantly reduced under the higher dose of desipramine, while the fluvoxamine acute treatment had no effect on any of the indices of timing. For chronic treatment, rats were injected intraperitoneally twice daily during the 29 days of the experiment. In this study, desipramine had a limited impact on timing behavior and no significant effects on timing indices, while chronic fluvoxamine treatment led to a reduction in the T-peak without affecting the WF. As desipramine has only minor affinity with SERT, while fluvoxamine mainly blocks the serotonin transporter, these results suggest the involvement of increased serotonergic signaling, caused by chronic fluvoxamine treatment to the decreased T-peak.

Bayley, Bentley, and Dawson [102] further investigated the effects of other selective SERT and NET inhibitors (imipramine, zimelidine, and clomipramine), as well as diazepam and d-amphetamine. D-amphetamine significantly increased lever-pressing rates, whereas all other compounds decreased it. The T-peak was largely unaltered by these compounds, indicating that timing accuracy remained relatively consistent. However, zimelidine at a high dose led to a significant reduction of the T-peak, accompanied by a decrease in spread time, pointing to an effect on interval timing. Again, among these drugs, only zimelidine was specific to SERT, and this result supports the reduction of the T-peak after SSRI treatment, but only for high doses of this specific SSRI and for longer fixed intervals (72 s in this study vs. 30 s in the previous one [100]). In comparison to Ho and colleagues, in this study, the effect was present after acute treatment. However, this interpretation should be taken with caution as a substantial decrease in the response rate was also observed in this condition.

Further, a comprehensive analysis of the effects of serotonin receptor agonists and their interactions with antagonist compounds on timing behavior in the PI procedure was provided [69]. Injection of the 5-HT1A receptor agonist 8-OH-DPAT resulted in a reduction of the T-peak and an increase in the WF. These effects were effectively antagonized by WAY-100635, indicating the involvement of 5-HT1A receptors in these temporal alterations. Similarly, the 5-HT2 receptor agonist DOI also induced a reduction in the T-peak and an increase in the WF. In this case, the 5-HT2A antagonist ketanserin effectively counteracted the reduction of the T-peak, highlighting the impact of 5-HT2A receptors. Thus, 5-HT1A and 5-HT2a receptors seem to similarly modulate internal time flow and affect timing precision.

MacDonald and Meck [95] investigated the effects of clozapine (an atypical antipsychotic and a 5-HT2A, 5-HT2B, and 5-HT2C antagonist) and haloperidol (a typical antipsychotic and inverse D2, D3, and D4 agonist) on interval timing in male Sprague-Dawley rats. Employing the PI procedure with durations of 10, 30, and 90 s and three corresponding response options, they employed a tri-peak procedure to assess interval timing across multiple time intervals. The tri-peak response function revealed a trimodal distribution centered on expected reinforcement times for each response option. Clozapine induced a leftward shift in the tri-peak function. Conversely, haloperidol led to a rightward shift in the tri-peak function. In a subsequent study, Buhusi and Meck [96] added an experimental condition with gaps into the presentation of stimuli. During these gaps, the stimulus light was turned off during the trial, aiming to reveal the impact of working memory on performance. In this study, clozapine administration also caused a leftward shift of response function, similar in standard PI trials and in those with gaps, pointing to alterations in interval timing regardless of the conditions.

Additionally, Heilbronner and Meck [97] introduced a modified version of this paradigm, the bi-peak procedure, offering a large reward (four pellets) for pressing the levers during the 10-s and 40-s target durations, along with an extra “defection” lever that offered a 20% probability of receiving a small but immediate reward (one pellet). The use of the “defection” lever allowed researchers to assess the impulsivity parameter, wherein the smaller but sooner option leads to trial completion and food reinforcement with a small probability. The authors analyzed the relationship between the probability of a defection response and the T-peak and found a negative correlation, suggesting a link between impulsivity and interval timing. However, a correlation was only found for the 10-s lever and not for the 40-s one. The authors suggested that such a discrepancy might be due to the low probabilities of long trials caused by the high percentage of defections. Despite such an association, the effect of fluoxetine in this study decreased the rate of responses with the “defection” lever and did not affect the T-peak parameter, suggesting distinctive mechanisms underlying timing and impulsivity. Notably, cocaine and methamphetamine, which were also used in this study, heightened early defection responses, indicating increased impulsivity, while timing parameters were hard to assess due to the very high probability of defection trials. In summary, the interaction between impulsivity and timing is hard to assess with this particular experimental paradigm, leading to a very high probability of defection trials (89% on average).

Recently, Shapiro and colleagues [70] conducted a series of experiments to investigate the role of 5-HT1A receptors in temporal memory processes. In this study, male Sprague-Dawley rats were trained to associate interoceptive states induced by saline and the 5-HT1A agonist 8-OH-DPAT with either 5 s or 20 s PI schedules. The group of rats trained to associate short intervals with 1A activation demonstrated a rightward shift of the T-peak, whereas the group of rats trained to associate long intervals with 1A activation demonstrated a leftward shift of the T-peak. The authors suggest that the administration of 8-OH-DPAT disrupted temporal memory retrieval, as reflected by peak functions consistently centered around 7 s, regardless of the trained interval. The rats displayed the ability to differentiate drug states and retrieve corresponding durations, yet a notable challenge emerged in responding accurately when the drug signaled the long duration. In a further experiment with the 5-HT1A antagonist WAY-100635, short or long intervals were conditioned with tones or light stimuli. A 5-HT1A blockade had no discernible impact on timing, although peak rates declined. The results imply that 5-HT1A receptor manipulations cause alterations in temporal memory and the pattern of response.

Although it implicated FI schedules, Dellu-Hagedorn’s study did not assess key parameters of time perception, only examining the total activity level, thus precluding any inference about time perception [15]. In particular, rats were divided into two groups: rats with a high number of lever presses were referred to as impulsive rats (IMP-FI) and rats with a lower number of lever presses were referred to as non-impulsive rats (NIMP-FI). The two groups did not differ significantly in activity level during the initial 10 s of the FI period, which was 60 s in total. However, IMP-FI rats displayed a notably higher level of activity until the delivery of food, resulting in more premature responses. Based on post-mortem neurochemical analysis, IMP-FI rats exhibited higher basal serotonergic turnovers in specific brain regions, including the prelimbic cortex, posterior insular cortex, hippocampus (HPC), and ventrolateral striatum (VLS). Lower serotonergic turnovers were observed in the dorsolateral orbitofrontal cortex (DLO) of IMP-FI rats. Positive correlations between anticipatory hyperactivity and serotonergic turnover were identified in the HPC and VLS regions. This study provides evidence for higher 5-HT metabolism in the HPC, prelimbic and posterior insular cortices, VLS, underlying impulsive behaviors based on the number of lever presses in the FI timing procedure. However, the relevance of these findings to interval timing is questionable.

Acosta and colleagues [123] conducted a series of experiments employing the PI schedule paradigm on mouse-specific models that were prenatally treated with valproic acid (VPA), which induces human-like autistic features in rodents. The results indicated that both female and male mice prenatally exposed to VPA exhibited diminished temporal accuracy and precision in interval timing tasks compared to their control counterparts. However, biochemical analysis revealed no significant differences between groups regarding 5-HT and 5-HIAA levels in the striatum, while the DA level was altered, suggesting that the effect on interval timing in this model was due to the functioning of the dopaminergic system.

Next, Miyazaki’s research [16] significantly advances our understanding of how serotonin regulates temporal perception, particularly in the context of waiting for rewards. Researchers confirmed the capability of optogenetic stimulation of the serotonergic neurons in the dorsal raphe nucleus (DRN) and its axon terminals in the selected brain areas. As verified through in vivo microdialysis experiments in various brain regions, optogenetic stimulation induced localized serotonin release in the orbitofrontal cortex (OFC), medial prefrontal cortex (mPFC), and nucleus accumbens (NAc). The study proceeded to examine the impact of serotonergic stimulation in distinct brain regions on waiting ability in experiments involving a sequential tone–food waiting task under varying reward delay conditions. During the test, mice were required to hold their noses in designated apertures while awaiting either a conditioned tone or a food reward. Following the tone cue, mice had to transition to the reward site and maintain their nose-poking behavior until the reward delivery. Two types of conditions were introduced: with certain reinforcement after fixed intervals of 6 or 10 s in separate conditions (certain condition) and uncertain conditions when reinforcement can occur either after 4, 6, or 8 s within a condition (or 2, 6, or 10 s in another condition). The findings revealed that the serotonin stimulation in the OFC promoted patience and significantly increased the T-peak. On the other hand, serotonin stimulation in the mPFC similarly affected the T-peak exclusively under uncertain future reward timings. Interestingly, the serotonin stimulation of the NAc region did not exhibit a significant impact on waiting behavior. This study revealed that sufficient serotonin levels in the OFC and mPFC regions promote non-impulsive behavior, particularly while waiting for the reward.

Summing up the research on the PI procedure, the predominant focus was on using monoamine microdialysis or serotonergic drugs as an exposure agent. At the same time, receptor regulation was not extensively covered, with only limited emphasis. Generally, most antidepressants and raphe nuclei lesions on rats did not affect timing abilities, except for a single instance where chronic fluvoxamine treatment reduced the T-peak. However, the administration of selective agonists for 5-HT1A and 5-HT2A receptors resulted in a reduction of the T-peak. These effects were effectively blocked when the corresponding receptor antagonists were administered, affirming the role of these receptors in mediating mechanisms of timing behavior. However, the atypical antipsychotic clozapine, the antagonist of 5-HT2A, 5-HT2B, and 5-HT2C receptors, also leads to a decrease in the T-peak, suggesting inconsistent effects of different serotonergic drugs in mediating interval timing in the FOPP. An optogenetic study further revealed that serotonergic stimulation of the OFC and mPFC regions increases the T-peak, promoting longer waiting times for rewards.

#### 3.4.4. Differential Reinforcement of Low Rate (DRL) Paradigm

The differential reinforcement of low rate (DRL) paradigm, also called the inter-response time greater than t schedule, is another experimental setup designed to assess temporal behavior. In this test, animals are trained to continuously respond, with the inter-response time (IRT) being longer than a standard preset time interval (t seconds), and such responses are reinforced. The major difference from the previously described task (PI) is the absence of the external presentation of time intervals. In relation to the FOPP, the task here seems easier as only one lever is available. Traditionally, this paradigm is widely used to examine the effects of antidepressant-like drugs, showing high predictive validity. Performance in this task is also clearly related to response inhibition as animals have to withhold from the response for a certain period of time. Thus, it can also measure impulsivity.

The following parameters are used to evaluate timing behavior in the DRL:

**Response Rate** (responses/min): This metric quantifies the average rate at which animals generate responses. It is calculated by dividing the total number of responses by the time of the session.

**Reinforcement Rate**: This metric reflects response efficiency and is calculated as the number of reinforced responses (those exceeding a standard preset time interval) during a session, divided by either the session time or the number of peak responses. An increase in this rate following antidepressant treatment is often referred to as an antidepressant-like effect.

**Peak Location** (PkL): This refers to the position (in seconds) of the highest point in the Inter-Response Time (IRT) distribution, reflecting the expected criterion time for reinforcement as determined by the rats.

**Peak Area** (PkA): This represents the area under the curve around the peak, indicating the concentration of responses near the peak time and reflecting the precision of responses over time. This measure also relates to peak spread and the coefficient of variation. Weber’s fraction, which provides a convenient measure of variability or precision in temporal differentiation, is derived by dividing the standard deviation of the response distribution by its mean, expressed as PkL.

A summary of the studies in which this paradigm is used is presented in Table 6.

The DRL procedure, with a typical standard preset time interval of 72 s, was extensively investigated in relation to antidepressant treatments. Studies from the 1980s to the 2000s were systematically reviewed by O’Donnell and colleagues [124]. To avoid redundancy, the reviewed articles are not included in our analysis. The review by O’Donnell and Seiden showed that many SSRIs, tricyclic antidepressants, atypical antidepressants, 5-HT2A antagonists, 5-HT1A agonists, monoamine oxidase inhibitors, and 5-hydroxytryptophan (5-HTP) were found to increase reinforcement rates while decreasing response rates. Conversely, amphetamines exhibited the opposite effect, decreasing response rates and increasing reinforcement rates. Non-selective 5-HT2 antagonists had no significant impact on these parameters, while benzodiazepines produced inconsistent results. Regarding the IRT distribution, it was observed that 5-HT2A antagonists, monoamine oxidase inhibitors, 5-Hydroxy-tryptophan (5-HTP), tricyclic antidepressants, and atypical antidepressants caused a rightward shift in PkL. In contrast, amphetamines caused a leftward shift in PkL. 5-HT1A agonists, non-selective 5-HT2 antagonists, SSRIs, and benzodiazepines did not affect the PkL shift but altered the shape of the IRT distribution.

Bradshaw and Szabadi’s research group used this paradigm on female Wistar rats to explore serotonergic regulation of time perception [66,67]. The standard time interval was equal to 15 s. In the first study, an experimental group received dorsal and medial raphe nuclei lesions using 5,7-DHT. The control group underwent sham lesions. The experimental group demonstrated more premature responses (decreased PkL), exhibiting heightened response rates. The IRT distribution in this group was significantly wider compared to the control group (increased PkA), which indicates that responses became less consistent. Supporting the role of the serotonergic system in this task, further neurochemical analysis revealed decreases in serotonin (5-HT) and 5-hydroxyindoleacetic acid (5-HIAA) levels for the experimental group, with no significant alterations in dopamine and noradrenaline levels [66]. A subsequent study validated and expanded these findings, showing that the impact on DRL schedule performance was independent of the reinforcement magnitude (sucrose concentration). Also the animals with greater food deprivation (lower weight) showed slight but significantly g higher PkL and reduced variability (lower PkA) compared to those of higher weight rats indicating the effect opposite for what can be excepted if the 5-HT disruption increase reinforcer value [67]. The results of these studies allowed researchers to assume that damage to the serotonergic pathways might impede animals’ ability to inhibit premature responses and impair their temporal discrimination skills.

Sokolowski and Seiden examined the effects of SSRIs (i.e., fluoxetine, sertraline, and paroxetine) on DRL (72 s) schedule performance [99]. The findings revealed distinctive alterations in behavioral patterns. Fluoxetine and sertraline significantly decreased response rates, while sertraline and paroxetine increased the PkL. At the same time, paroxetine induced a decrease in the PkA. Since different drugs had different effects, it could be due to different binding constants. However, the authors assume that such effects may be a consequence of a more complex interplay of factors, such as interactions with other receptors.

In addition, there is evidence of the influence of amphetamine on DRL performance. High-dose methamphetamine treatment was found to damage both dopamine and serotonin neurons, resulting in significantly reduced serotonin concentrations in various brain regions [103]. These treatment decreased PkA but did not change PkL, suggesting increased variability in responses and a less precise representation of duration.A further study aimed to extend the operant model of the DRL 72 s schedule to mice through genetic and pharmacological interventions [91]. To compare the effects of drug administration on behavioral performance, the authors implemented treatments with serotonergic substances on C57BL/6J mice wild-type (WT) and 5-HT1AR knock-out mice (5-HT1AR KO). Analysis of baseline activity showed no significant difference between these two groups before or after drug administration. The acute administration of fluoxetine demonstrated an antidepressant-like effect in 5-HT1AR KO mice, marked by a significant decrease in response rate and an increase in PkL. However, these effects were not statistically significant in WT mice. When fluoxetine was co-administered with the 1A antagonist WAY-100365, there was no change in the response rate for 5-HT1AR KO mice, but a significant decrease was observed for the WT group. Analysis of IRT distribution revealed a significant increase in PkA and PkL for both WT and 5-HT1AR KO mice. Desipramine, administered to both 5-HT1AR KO and WT mice, led to decreased response rates and a significant increase in PkA and PkL, which was again evaluated as an antidepressant-like effect. In summary, the results indicate that fluoxetine induced a significant increase in PkL only when serotonin 1A receptor activity was suppressed, as seen in 5-HT1AR KO mice or when co-administered with a 1A antagonist in the WT group. Blocking 1A receptors seemed to potentiate the antidepressant-like effects on timing abilities. Overall, through genetic and pharmacological approaches, the study validates the extension of the DRL 72 s operant schedule as a behavioral model for antidepressant action in mice, which could also serve as a valuable tool for further interval timing studies.

Recently, two studies assessed the potential antidepressant-like effects of psychedelics by comparing them with the effect of fluoxetine on inter-response timing in the DRL 72 s schedule [82,98]. As predicted, fluoxetine reduced response rates and increased PkL, consequently enhancing the reinforcement rate. Similar effects were observed with (S)-ketamine but not (R)-ketamine. Further, psilocybin administration demonstrated an immediate antidepressant-like effect, characterized by increased reinforced presses, an improved response efficiency, and an increase in PkL [82]. However, these effects were not sustained in the post-treatment phase (weeks 2–5). In contrast, LSD had no statistically significant effects on behavior in this task. These studies demonstrate that similar effects in this paradigm, including the shift in PkL, associated with interval timing, can be observed both with drugs with known serotonergic involvement as well as with (S)-ketamine, which might not affect serotonergic functioning. At the same time, the effects of drugs with known influences on serotonergic activity are rather consistent with increased serotonergic activation associated with an increase in PkL.

In summary, a consistent relationship between the level of serotonergic activity and timing parameters in the DRL schedule is observed with decreased activity caused by the destruction of raphe nuclei or 5-HT receptor antagonists, leading to decreases in PkL and reinforcement rates, while increased reinforcement rates and increased PkL are associated with the increased serotonergic functioning induced by different drugs. In particular, the activation of 5-HT2A receptors and SERT inhibition increased the PkL parameter, and the 5-HT1A receptor blockade potentiated the effects of SSRIs.

#### 3.4.5. Delay Discounting Task

Delay discounting (DD), also known as temporal discounting, is a behavioral test used as a common method for studying impulsivity and decision-making. The key measure of the paradigm reflects an individual’s tendency to choose smaller but sooner (SS) or larger but later (LL) rewards. The relation between the subjective value of a reinforcer and the delay until its receipt can be described by a discounting function. Such behavior may be based on the peculiarities of time perception [125,126,127]. In situations requiring a waiting period, impulsive behavior, as an increase in SS reward preference, can be provoked by a tendency to overestimate forthcoming time intervals. The following parameters can be used to assess parameters linked to timing perception in this paradigm:

**Preference of the LL reward** (%LL): The ratio of LL choices to the total sum of all reinforced choices. This parameter reflects the preference for delayed but larger rewards (LL) in contrast to smaller, immediate rewards (SS).

**Number of omission trials** (K_0_): The count of omission trials, serving as a measure of engagement and the animal’s training level. Higher engagement and a well-trained animal are often associated with a lower number of omission trials.

**Degree of delay discounting**: The rate at which the subjective value of a reward decreases as the delay in receiving the reward increases. A steep rate of discounting results in smaller values of larger delayed rewards and, therefore, increased choices of the more immediate small rewards.

Among the reviewed studies, one article applied the test on chicks, one on mice, and six on rats (Table 7). Other previous studies were not found among the screened literature, though a short summary of these results by 2012 is provided from the review of Homberg [128]. Generally, results from psychopharmacological studies suggest a link between increased serotonergic functioning and decreased delay discounting.

Matsunami and colleagues [101] investigated the behavioral effects of an SSRI, fluvoxamine, using domestic chicks in an I-shaped maze with a feeder. The study focused on intertemporal choices, presenting chicks with two colored beads linked to SS or LL rewards. Fluvoxamine administration resulted in a suppression of the preference for the SS option. This suggests that increased 5-HT levels made chicks more tolerant of delayed food rewards, which underscores its role in enhancing patience and reducing impulsiveness. This finding was corroborated by in vivo microdialysis experiments, which demonstrated that fluvoxamine led to heightened extracellular levels of both 5-HT and DA in the medial striatum region. The authors proposed a neuromodulatory mechanism involving both neurotransmitters in the regulation of decision-making.

Several studies examined DD using rats that underwent treatments with selective 5-HT ligands. Zaichenko and colleagues [71] investigated the effects of a 5-HT1A serotonin receptor agonist and antagonist on rat behavior in a situation with free choice. Based on the preference of food reinforcement, rats were categorized into two groups: preferring delayed but more valuable reinforcement (self-controlled) and preferring immediate but less valuable reinforcement (impulsive). One hour prior to testing, the rats were intraperitoneally treated with the 5-HT1A receptor agonist 8-OH-DPAT or the selective 5-HT1A antagonist WAY-100635. The study assessed various parameters including pedal presses, time of nose pokes to the magazine (latent period), and missed responses. No effect of WAY-100635 was significant but 8-OH-DPAT, while also increasing omission responses in both groups, specifically increased the proportion of LL reinforcements in impulsive rats. Next, Zaichenko and colleagues [80] investigated the effects of the 5-HT2 receptor agonist DOI and the antagonist ketanserin on rat behavior. The administration of ketanserin resulted in a significant increase in the selection of SS reinforcement in the impulsive group, while it had no effect on the self-controlled group. DOI demonstrated significant reductions in impulsivity across all animals and increased LL reward preference. The results of these studies supported the involvement of 5-HT1A and 5-HT2 receptors in controlling impulsive behavior, particularly in the context of reinforcement value selection. The studies highlighted that the impulsive animals may be more sensitive to alterations in serotonergic system activity compared to the self-controlled animals, demonstrating increased LL reinforcement preference under 5-HT1A receptor activation and increased SS reinforcement preference under a 5-HT2A receptor blockade.

Kirkpatrick and colleagues [129] investigated the influence of early-rearing conditions on impulsive decision-making in rats. Rats were divided into two groups: reared in isolation (IC) and reared in an enriched environment (EC). Neurochemical analyses were conducted to examine monoamine concentrations in specific brain regions (prefrontal cortex and nucleus accumbens) and their correlation with behavior in different tasks. Results revealed that the IC group exhibited a higher level of impulsivity compared to the EC group, indicating the significant influence of early social and environmental factors on impulsive decision-making. Moreover, concentrations of NE and 5-HIAA in the NAc were positively correlated with LL reward preference, highlighting the involvement of these regions in decision-making processes.

Buhusi and colleagues [90] studied stress-induced impulsive choice behavior in a mouse model of brain disorders. Close homolog of L1 (CHL1) gene dysregulation is associated with schizophrenia, so CHL1-deficient mice are utilized as a model of schizophrenia-like deficits, including interval timing and spatial memory impairments. The authors employed a DD paradigm with varying time delays (0, 4, 16, and 64 s) for LL rewards in wild-type (WT) and CHL1 knock-out mice (CHL1-KO). Initially, animals underwent a baseline condition for training, followed by a 21-day exposure to chronic unpredictable stress. After exposure to the stressing conditions, CHL1-KO mice showed increased impulsive choices of SS reinforcement. Then, two separate groups were formed in order to conduct the cFos analysis and further evaluate the effects of drug treatment on stressed animals. The first group (CHL1-KO n = 6; WT n = 6) underwent cFos immunostaining, revealing decreased neuronal activation in stressed CHL1-KO mice compared to stressed WT animals, specifically in the prelimbic cortex and dorsal striatum. Notably, this decrease was not observed in other regions such as the orbitofrontal cortex and the core and shell of the NAc, which are typically associated with valuation processes. The second group (CHL1-KO n = 10; WT n = 10) was treated with the 5-HT2C agonist Ro60-0175, exhibiting a reversing effect on impulsivity (i.e., an increased preference for the LL reward), which was observed in both stressed CHL1-KO mice and the WT controls. While the effects of Ro60-0175 on DD were not specific to genotype, the results suggest that targeting the 5-HT2C receptor could be a valuable strategy for treating disorders characterized by impulsivity and other possible timing behavior impairments.

The study by Dellu-Hagedorn and colleagues is more relevant here, but it was also touched on in Section 3.4.3 [15]. The rats were categorized into two groups based on %LL, so impulsive rats had a low preference for the LL reward (%LL < 35%), and non-impulsive rats had a high preference for the large reward (%LL > 53%). Impulsive rats exhibited lower basal turnover rates of both serotonin and dopamine in critical brain regions, including the amygdala central nucleus, ventrolateral striatum, and NAc core, compared to their non-impulsive counterparts.

Jiang and colleagues [79] conducted a study to investigate the effects of selective ligands on prospective decision-making in rats. The selective 2A agonist DOI was found to decrease LL reward preference, increasing impulsive decision-making. Notably, the effects induced by DOI were specifically blocked by the 5-HT2A receptor antagonist ketanserin, but not by the 5-HT2C receptor antagonist SB-242084. In contrast, the non-selective monoaminergic drug lisuride decreased impulsive decision-making. The effects of lisuride were not antagonized by ketanserin, the selective 5-HT1A antagonist WAY-100635, or the selective dopamine D4 receptor antagonist L-745870, but were attenuated by the selective dopamine D2/D3 receptor antagonist tiapride. The study concludes that DOI and lisuride have opposing effects on impulsive decision-making, acting via distinct receptors. The DOI-induced increase in impulsivity is mediated by the 5-HT2A receptor, while the lisuride-induced inhibition of impulsivity is apparently regulated by the dopamine D2/D3 receptor. Additionally, another study with psilocybin treatment, a 2A agonist, had no significant effects on decision-making parameters in delay discounting tasks in both male and female rats [84].

The dynamic of the 5-HT1A and 5-HT2 receptor, SERT, and DA regulation processes underlying impulsive behavior emphasizes the correlation between the neurochemistry of brain regions and impulsive decision-making. However, the effect of DOI, an agonist of the 5-HT2A receptor, was inconsistent across studies as both increases and decreases in %LL were reported. At the same time, lower 5-HT and DA levels within the striatum, NAc, and amygdala lead to heightened impulsivity and a preference for SS reinforcement. On the other hand, activation of 5-HT1A and SERT reduced impulsiveness in animals, increasing LL reinforcement choices. The identification of distinct serotonin and dopamine metabolism patterns associated with impulsive choices holds promise for a deeper understanding of the connection between mental disorders and potential therapeutic interventions targeting monoaminergic pathways.

#### 3.4.6. Other Behavioral Tests

Interval timing in animals is studied with a wide range of experimental paradigms, and each of them offers unique insights. In this section, we would like to discuss several studies that indirectly touch on the problem of time perception mechanisms by applying other experimental paradigms. By scrutinizing reward-seeking behavior, studies also indirectly touch on the intricate relationship between time perception and physiological states. Two other paradigms constitute a rich landscape for exploring the multifaceted dimensions of timing behavior in animals, opening avenues for a comprehensive understanding of the neural and pharmacological underpinnings of temporal processing (Table 8).

In a study involving two male rhesus monkeys performing a modified Go/No-Go task, the dorsal raphe nuclei were found to play a crucial role in encoding information related to temporal aspects. Neurons within the dorsal raphe nuclei were observed to encode various temporal elements, including schedule onset, reward expectation, outcome, and the amount of reinforcement [130]. A notable subset of these neurons, termed “reward expectation” neurons, exhibited increased firing rates as the monkeys awaited the delivery of the reward. This pattern of neural activity was interpreted as encoding the prediction of the reward outcome, suggesting a link to the perception of time intervals. Moreover, a distinct subset labeled as “schedule onset” neurons displayed specific activity patterns, dependent on whether a trial was the first in a schedule or not. This activity aligns with a framework in which the raphe nuclei coordinate persistent goal-seeking behavior, processing information about progress throughout the task and mediating the temporal pattern of the responses. While time perception was not the primary focus, it was demonstrated that the modified Go/No-Go task holds significant promise for understanding the neural mechanisms of timing, especially in relation to the neural activity mediated by the serotonergic system.

Recently, McLaughlin and colleagues [92] conducted a study investigating the interplay between cannabinoid receptor 1 (CB1) ligands and 5-HT1A receptor blockades in mediating impulsivity, considering timing mechanisms in rats. The researchers employed the fixed consecutive number (FCN) task, a paradigm designed to evaluate the tendency of rats to prematurely interrupt a series of responses. In this task, rats were trained to press the ‘counting’ lever a preset number of times, and reinforcement was provided upon switching to the ‘reinforcing’ lever after the required presses. As a result, the CB1 inverse agonist AM251 and the CB1 antagonist AM6527 were found to induce more premature switches to the ‘reinforcing lever’, but only in the presence of the 5-HT1A antagonist WAY-100635. However, the peripherally restricted antagonist AM6545 did not induce impulsivity, regardless of the 5-HT1A blockade. Crucially, it was noted that when performing the FCN task, animals apparently relied upon timing mechanisms rather than counting [131]; therefore, the authors incorporated a variable inter-response interval with the ability to press a response only after a discriminative signal (a variable consecutive number task with a discriminative stimulus) to eliminate the influence of timing behavior on performance. In this version of the task, substantial impairments were found for only the combination of WAY-100635 and AM251, implying that the effects of the antagonist AM6527, together with the 5-HT1A blockade seen in the FCN task, could be due to the influence of timing alterations. This study serves as an example of how integrating timing aspects could enhance the depth of research in impulsivity and other behavioral domains.

In conclusion, studying reward-seeking behavior by considering aspects of time perception can reveal and elucidate the internal mechanisms of behavior and decision-making. It was shown that the raphe nuclei neurons play a key role in processing information about the task’s progress and mediating temporal response patterns, highly contributing to the maintenance of goal-seeking behavior. Another study has shown that 5-HT1A receptors are involved in the timing alterations induced by specific CB1 antagonism, since co-administration of the 5-HT1A antagonist impaired task performance when temporal aspects of the task were incorporated into the task for impulsivity. This study exemplifies how integrating timing aspects can deepen research in impulsivity and various behavioral domains. The multifaceted dimensions of timing behavior explored in these studies underscore the need for further comprehensive investigations into the neural and molecular mechanisms of temporal processing in animals, particularly concerning serotonergic system mediation.

## 4. Discussion

This systematic review examines the involvement of the serotonergic system in time perception across species. Overall, 50 animal and 10 human studies were reviewed in the Results Section, all of which investigated evidence from a broad spectrum of tests designed to assess various facets of time perception on a scale of several seconds. The reviewed studies applied 11 main different experimental paradigms that largely differed between human and animal studies. Thus, we have come to the major challenge of this review, which is to link human and animal studies together by relating them to general concepts of time perception, such as internal time speed. Here, we chose to interpret the results according to the dual klepsydra model (DKM). However, while other theories might also account for the results, here we did not aim to compare different theories but to provide an integral view of the reviewed results, which DKM allows. DKM postulates that internal time is related to external information inflow and the discharge rate of the lossy neuronal accumulator for time [17,112]. Thus, we relate an internal time deceleration to a slower discharge rate, as well as an internal time acceleration to a faster discharge rate of the neuronal accumulator for time.

Overall, we assessed 11 experimental tasks that were further grouped into retrospective, immediate, and prospective paradigms based on whether the reference point for the response to be provided was centered around a past interval, ongoing interval, or anticipated duration to obtain reinforcement. While this distinction seems relevant, there are many other nuances in experimental paradigms that influence temporal behavior and should also be considered, such as the duration of the interval to be compared. While we specifically focused on durations within several seconds, one might also suggest that coding durations of up to 5 s might be much easier than coding durations of 72 s, even if the same mechanisms are in place. Moreover, performing a task that requires such a long interval might put an additional burden on other cognitive functions, such as impulsivity, inhibitory control, and motivation. However, there was no evident discrepancy between results obtained from shorter and longer time ranges.

Another source of evidence, which is difficult to fit into the classification used, is psychological questionnaires. Although such a method possesses limitations and does not always provide information regarding the direction of time perception changes, it bridges the gap between psychophysical and introspective evidence [132]. For instance, with Jasper’s questionnaire of time passage subjective speed, Portnova and colleagues [106] demonstrated that higher serotonin function (MAOA variants causing less efficient degradation of serotonin and 5-HT2A receptor variants associated with increased receptor density) was associated with subjectively prolonged time passages, which is characteristic of accelerated internal times. Other studies reported disturbances in subjective time associated with heightened serotonergic function [85,107]. Nonetheless, direct linkage of this subjective feeling with psychophysical experiments is missing and might be a possible direction for future research.

### 4.1. 5-HT Modulation of Retrospective Timing

Cumulatively, human studies using retrospective timing tasks showed that more efficient serotonergic neurotransmission is associated with an increased speed of internal time. In a study using the duration discrimination task (DDT) and genetic analysis of serotonergic neurotransmission, it was shown that carriers of genotypes with predisposed higher serotonin neurotransmission (polymorphisms coding lower 5-HT reuptake, lower 5-HT degradation, and higher 5-HT2a receptor density) exhibit increased internal time speed according to psychophysical parameters (decreased PSE and increased temporal discharge rate, K, based on greater subjective shortening of the first compared to the second durations presented in a pair) [111]. Additional evidence was obtained using clinical samples. For example, a study that recruited a sample with clinical depression showed increased PSE values and increased iAEP, a neurophysiological marker of central serotonergic activity, in this group compared to controls. This finding suggests that less efficient serotonergic neurotransmission in depression is accompanied by a decelerated internal time speed [113]. The opposite effect was shown for patients with schizophrenia, who tended to overestimate time intervals and had decreased iAEP values [107]. Overestimation of duration can be interpreted as increased internal time speed/temporal accumulator discharge, considering that current duration is compared to some predefined standard given by society, and this predefined duration representation could be less affected by serotonin. Thus, this finding supports the link between increased serotonergic transmission and higher internal time speed.

Previous research and observations are generally consistent with this interpretation. In schizophrenia, time perception disturbances are often noted by clinicians, and there is some experimental evidence [107,133]. Several studies report an overestimation of intervals, which indicates an accelerated internal time speed [134,135], and some hypothesize that this acceleration might be related to positive symptomatology [135]. However, a meta-analysis conducted by Thoenes and Oberfeld [134] reports that time perception in schizophrenia is consistently characterized by the loss of precision. At the same time, there were no consistent findings regarding the direction of the shift in accuracy towards an accelerated or decelerated internal time speed. According to the serotonin hypothesis of schizophrenia, time perception deviations might be caused by the chronic serotonergic overdrive observed in this disorder [108,110]. Drugs with antagonistic effects upon 5-HT2A receptors are one of the main ways to treat schizophrenia, addressing positive symptomatology [109]. Nonetheless, it is important to bear in mind the vast number of potential confounders in time perception studies in schizophrenia, such as overall working memory deficits [136].

As for depression, clinical observations and self-reports of subjective time disturbances indicate time passing more slowly, although one meta-analysis reported that experimental evidence is inconsistent [137]. Although the evidence obtained from clinical samples should be considered cautiously, it indicates an effect in the same direction as experimental evidence from the neurotypical samples reviewed above. The serotonergic system is implicated in the etiology of depression and is considered to be less efficient than in neurotypical people [138,139].

In general, time perception studies using clinical samples can be insightful; however, they contain many pitfalls concerning the problem of the pathogenesis of psychiatric disorders, which is still unresolved, and alterations to the serotonergic system are probably accompanied by other neurotransmission changes. Therefore, studies on animal models are crucial to reveal the exact neurobiological causes of the time perception deficits observed in these disorders.

While human studies primarily focus on the general and indirect influences of the 5-HT system, animal studies provide a more nuanced evaluation of serotonergic neurotransmission. For example, sources of evidence include manipulation of the raphe nuclei and its projections, the effects of psychoactive substances, and the involvement of specific receptors.

Moving to the interpretation of animal data, we need to point out one important difference—we infer animals’ time perception based on their responses in specifically designed experimental situations. In most cases, animals must undertake extensive training to learn the time intervals and adjust their behavior to it. Only after this period the experimental manipulations that affect serotonergic functioning usually are introduced. This type of design makes these tests more comparable to duration estimation (DE) or duration production (DP) tasks in humans, where the current duration is compared to some previously learned duration representation or standard. While this standard is given in conventional units for humans, such as seconds or minutes, for animals, it can be stored in the form of duration representations related to particular actions. Thus, seemingly contradictory findings on the opposite shift between the bisection point (D50) in the interval bisection task (IBT) in animals and PSE in DDT in humans can be explained if we take this aspect into account. In particular, while PSE decreases with increases in internal time speed, operationalized as an increased temporal discharge rate (K), on the contrary, D50 increases, as more objective time has to pass for the duration to be perceived as similar to the duration standard established in the baseline condition, because only the current duration shrinks more with a greater discharge rate.

Considering the above-described logic, data from the animal IBT paradigm can be interpreted as a direct link between serotonergic functioning and the internal time discharge rate. For instance, D50 consistently decreases with the destruction of the raphe nuclei, the primary source of serotonin projections [56,57,59], and with the blockade of 5-HT 2C receptors, suggesting that the internal time speed decreases with the elimination of serotonergic functioning. At the same time, substances can cause an increase in serotonergic functioning. For example, SSRIs and agonists of 5HT2A receptors, lead to an increase in D50, interpreted as an increase in the internal time speed.

Thus, human and animal studies logically converge on the general link between internal time speed, i.e., the discharge rate of the temporal neural accumulator, and serotonergic functioning, with SERT, 5-HT2A and 5-HT2C receptors being implicated in this process.

### 4.2. 5-HT Modulation of Immediate Timing

Using immediate timing tasks in the framework of the DKM, human studies also demonstrated that the more active the serotonergic neurotransmission, the faster the internal time speed. In the duration reproduction task (DR), the higher speed of the neural accumulator discharge can be observed as a trend towards under-reproduction of the intervals. Specifically, psilocybin treatment, which primarily acts as a 5HT2A receptor agonist, induced a higher loss rate of the neural accumulator, i.e., accelerated internal time [85,86]. The only human study that is inconsistent with the link between increased serotonergic functioning and increased internal time discharge is on the effect of LSD on the DR task [81]. This inconsistency might be attributed to the intricate relationship between the activity of the neuromodulatory systems evoked by the substance.

In animal studies, this difference was noted when comparing the action of LSD and psilocybin in the DRL test, as only psilocybin affected PkL and not LSD [82]. Regarding humans, PET studies have shown that the 5-HT2A receptors are mainly responsible for the psychedelic effects of psilocybin [140]. In addition, receptor-enriched analysis of fMRI functional connectivity (FC) showed distinct phenomenology and connectivity patterns of the effects of LSD on the serotonergic and dopaminergic systems [141]. D1- and D2-enriched FC within the insular, opercular, motor, and superior parietal regions induced by LSD in comparison to the placebo correlated with disembodiment and impaired cognitive control. At the same time, serotonergic system FC induced by LSD correlated specifically with perceptual disturbances. Such an approach opens new avenues for time perception studies in humans due to the possibility of functional mapping the activity of neuromodulatory systems during experimental tasks. However, the exact mechanisms of neuromodulatory activity induced by LSD are still unresolved.

Another important point to mention is the high individuality of time perception. In the study by Yanakieva and colleagues [81], this aspect was neglected as the between-subject design was applied. Therefore, given the unaddressed individuality of time perception and the inconsistency of LSD’s effect on neuromodulation, we suppose that the influence of LSD on time perception is still to be scrutinized.

Nonetheless, taken together and considering the overall quality assessment score as a factor in interpretation, human studies using immediate paradigms point to accelerated internal time with more active serotonergic neurotransmission, which is primarily mediated by the 5HT2A receptors. Unfortunately, the studies found do not provide more nuanced details on the mechanism of the serotonin system’s effect on time perception. On the other hand, experiments on animals, employing immediate timing paradigms such as the FOPP, DRL, and PI, provided a rich understanding of the involvement of different brain regions, projections, and psychoactive substances in time perception.

Contrary to the previous section on retrospective tasks, the results of the three experimental paradigms presented here seem hard to integrate. Results on DRL generally mirror those obtained in the IBT, with higher PkL and D50 values associated with increased serotonergic functioning by SSRI and 5HT2A agonists, while lower PkL and D50 values are observed with serotonin disruption (see Table 3 and Table 6). For the IBT, we linked increased D50 with increased internal time speed discharge, referring to the fact that the training phase was performed under baseline conditions and postulating that the learned duration representation on which the task performance is based is not affected by the experimental manipulation. Similar logic can be applied to DRL. While the response in DRL is based on the currently perceived duration, the animal has to relate it to the representation formed during the training phase to provide the correct response. Thus, decreased PkL, which is also sometimes called a “premature response”, can be interpreted as a decrease in internal time speed/a decrease in the discharge rate of the temporal accumulator.

The results on the T-peak shift in the PI schedule seem to be quite inconsistent among studies. However, most recent research supports the link between increased serotonergic functioning and T-peak increases. For example, optogenetic stimulation of serotonergic axon terminals in the striatum, OFC, and mPFC lead to T-peak increases [16]. In the PI tests, acute antidepressant treatment generally did not directly alter timing parameters. Chronic fluvoxamine treatment resulted in a leftward shift of the T-peak [100]. While the acute effect of T-peak decreases should be interpreted as decreases in internal time speed and therefore linked to serotonergic dysfunction, the chronic effect can be interpreted in the opposite way, as recalibration of the formed standard might be achieved. Consistent with this view, the atypical antipsychotic clozapine, known as a 5-HT receptor antagonist, blocking 5-HT transmission, reduced the T-peak under the acute treatment condition [95,96]. Thus, the results of the PI tests also might be reconciled with the link between increased serotonergic functioning and a higher internal time speed.

Findings in the FOPP schedule generally suggest that increased serotonergic neurotransmission, mainly caused by agonists of 5-HT2A and 1A receptors, leads to decreases in T50. However, this effect might be augmented by the blockade of 5-HT2C receptors (see Table 4). This effect is opposite to what is seen for PkL in DRL. In addition, this is observed, albeit in a less consistent way, for T-peaks in the PI procedure. The main difference between these tasks is that there is no requirement to withhold any responses in the FOPP: in the FOPP, animals have to choose between the levers to respond depending on the duration passed, with one lever response reinforced in the first half of the trial and another lever in the last part, while in both PI and DRL, only one lever is generally available and animals have to withhold from pressing it for a particular time to obtain the reinforcement. We can speculate that the ability to respond to any lever and adjust responses depending on the provided reinforcement might change the way the memory representation of duration is formed, and in contrast to DRL, PI, and the IBT, the recalibration of this duration representation might occur more quickly within the testing procedure. Thus, the increased interval time speed caused by serotonergic functioning can also explain this effect by promoting a shift of long-lever presses towards earlier values.

Overall, most of the reviewed studies can be reconciled with generally increased serotonergic transmission, particularly through 5-HT2A receptor activation, and accelerated internal time by increasing neural accumulator discharge rates. Sometimes, seemingly opposite effects can be reconciled by taking into account the details of the paradigm and the specific effects of drugs.

### 4.3. 5-HT Modulation of Prospective Timing

Another point to mention before discussing the results of animal studies is the link between impulsivity and time perception. Impulsivity is a behavioral trait that is closely linked to time perception [129]. While several cognitive processes contribute to impulsivity, a recent human study suggests the crucial role of interval timing in the performance of the delay discounting task, but not inhibitory control [142,143].

Prospective timing tasks may shed light on the intrinsic link between time perception and impulsivity in their modulation by serotonergic neurotransmission. The major experimental paradigm in this regard is DD. In such tasks, impulsivity manifests itself with a preference for smaller but sooner (SS) rewards over larger and later (LL) rewards.

In humans, the majority of DD studies imply decision-making regarding concepts of long periods of time, such as days or even months and years, which are beyond the scope of the present review and the time scale within which time perception is defined. Very few studies implement intervals within the range of the present time (from several seconds to minutes). Existing studies of this type have reported inconsistent findings on whether the discounting parameters relate to the internal time speed in terms of the direction of influence, although some links between these cognitive processes have been suggested [142,143].

In the present review, we managed to find two human studies that employed DD with intervals within the scale of several seconds. One study revealed that an increase in central serotonin levels caused by tryptophan supplements did not cause any changes in preferences for SS or LL rewards. However, the study found that serotonin levels modulated the activity of the dorsal and ventral striatum in anticipation of a reward. In particular, the activity of the ventral striatum was specific to SS reward anticipation, being enhanced when serotonin levels were high. At the same time, the activity of the dorsal part was specific to LL reward anticipation, increasing with higher serotonin levels [105]. Therefore, the study implies that there is some link between serotonergic neurotransmission and anticipation of immediate or delayed rewards. The absence of behavioral findings could be due to the insufficient motivation of the participants, as we suppose that drops of a sugary drink might not be an ideal reinforcement for human participants. Nonetheless, in the subsequent study, the experimental paradigm was modified, and the reward was monetary. With these modifications, researchers showed that decreased serotonin levels were associated with SS reward preference [104]. Thus, it could be suggested that with decreased serotonin levels, internal time decelerates, and it becomes more motivationally demanding to opt for LL rewards.

Due to the lack of human studies on serotonin modulation of behavior in prospective timing tasks, animal studies could help to clarify the issue. In animals, the DD paradigm sheds light on the dynamic regulation processes involving 5-HT1A and 5-HT2 receptors, as well as SERT. The examination encompasses the influence of brain regions, the effects of psychoactive substances, and the roles of specific receptors in shaping decision-making regarding delayed rewards. In a study on mice with schizophrenia-like deficits, chronic stress increased SS reward preferences. These behavior changes were followed by a decreased number of cFos-positive neurons in the prelimbic cortex and dorsal striatum [90]. Further investigation with a 5-HT2C agonist reversed the effect of impulsivity, increasing preference for LL rewards, suggesting a role of this type of receptor in promoting waiting behavior. SSRI and 5-HT1A receptor agonists also reduced impulsivity (see Table 7). However, substances that affect 5-HT2A receptors showed inconsistent effects on DD performance. The study of Kirkpatrick and colleagues [129] revealed that the concentration of 5-HIAA in the NAc positively correlates with the degree of delay discounting, reflecting a higher LL reward preference even with long intervals. Lower turnover rates of both 5-HT and DA in the amygdala central nucleus, ventrolateral striatum, and the core of the NAc contribute to heightened impulsivity and a preference for SS rewards [15]. Thus, while serotonin activation leads to decreases in impulsive choices and SS reward preference in this task, the effect might be more related to SERT and the involvement of 5-HT1A and 2C receptors.

The interpretation of the results in terms of internal time speed can be challenging. On the one hand, premature responses and SS reward preferences could be associated with an increased internal time speed. On the other hand, this contradicts the previously mentioned link between lower impulsivity and increased serotonergic activity, which we interpret as an increased internal time speed. Moreover, considering DRL and some PI studies, where, in general, serotonergic activity was associated with the ability to withdraw from the response for a longer time, increases in PkL and the T-peak were interpreted as an increase in internal time speed, as the ongoing duration processing is compared to the unaltered duration representation built from previous training experiences. The same logic can be applied here as the decision about the reward choice is based on the comparison of the ongoing perception with pre-established duration representation that is extrapolated into the future. Therefore, a quicker internal time speed leads to the perception of intervals as passing faster than they do objectively, thereby making it less motivationally demanding to opt for a delayed reward and reducing impulsivity.

### 4.4. Limitations and Future Directions

While we strived to provide an objective representation of the contemporary results on the link between interval timing and serotonergic functioning, it is important to acknowledge some limitations. First, the search criteria may not have encompassed all synonyms and related terms for “time perception”. This resulted in the identification of relevant concepts, such as “waiting” [16], “reinforcement” [73,144,145], and “temporal processing” [75,83,86,90,95,123,146,147] after the initial screening process, through a supplementary “snowball” search. While PubMed, APA PsycINFO, and APA PsycARTICLES provide extensive coverage of the biomedical and psychological literature, they may not fully encompass studies from certain interdisciplinary domains. To address this, we supplemented our database searches with hand-searching and citation tracking to identify additional relevant studies. However, including broader interdisciplinary databases, such as Scopus or Web of Science, might have captured additional studies from fields adjacent to our primary focus. Future reviews could consider integrating such databases to further enhance comprehensiveness.

Second, certain experiments evaluating impulsivity or non-timed anticipation have been excluded due to a predefined time scale ranging from seconds to minutes. This range was applied to studies on both animals and humans. While this approach provides a clear framework, it should be noted that different cognitive mechanisms may appear within this scale in humans and non-human animals, and direct translation of evidence between species might possess limitations. However, we should also acknowledge that within the current time scale, we did not find a clear difference in the effects described for relatively short (<10 s) and long (>70 s) intervals.

Third, while most of the studies included in the review were well-documented, there were some noteworthy considerations. Specifically, some studies employed samples limited to a certain sex, age group, or housing condition, which may have drawn a bias. Thus, for enhanced comprehensiveness and generalizability, it is recommended to incorporate additional controls for these factors in future research [114,115,148,149,150,151].

Additionally, in the current review, we provide integration of the results within the DKM framework. Although it seems to reconcile many different studies, there are certainly many unresolved questions and unproven assumptions that should be studied further. While we managed to draw a general conclusion, there are discrepancies regarding the scale of particular receptor types and implicated structures and how their involvement influences time perception. Our task was complicated by several factors. First, the 5HT neural system projects widely to many forebrain areas. Second, there are many different 5HT receptors, mostly post-synaptic but also some important presynaptic receptors that potentially modulate neuronal firing in opposite directions in these various forebrain areas.

The results of our review regarding the differential involvement of serotonergic receptors to the interval timing in particular paradigms should be considered with caution as this might be due to the uneven distribution of studies targeting different serotonergic receptors among paradigms, as could be seen in Table 1. It is also important to mention that serotonergic neurotransmission is tightly connected with other neurotransmitter systems, which may induce a chain of mutual influences and mixed effects on timing behavior [152,153,154]. For instance, dopaminergic neurotransmission also plays a crucial role in temporal cognition, which is discussed in a number of comprehensive reviews [20,155,156]. Moreover, many substances used to manipulate serotonergic neurotransmission often induce widespread and mixed effects involving different serotonin receptor types and even those of other neurotransmitters.

Therefore, further research into the effects of serotonergic substances, along with a deeper analysis of the interactions between different brain regions and neuromodulatory systems, is needed to provide valuable insights into the neural mechanisms of time perception. For instance, this could be realized using optogenetics. Another promising approach is developing novel and more selective drugs that allow us to better understand the complex nature of serotonergic modulation. From the behavioral side, there is also a clear need for a better description of cognitive functioning in animals, digging into nuanced aspects of interval timing implicated in different tasks. There is great potential in using automatized behavioral testing within ecologically valid conditions (e.g., with IntelliCage [157]). In addition, there is a need for detailed 5-HT projection mapping in both human [158,159,160] and animal brain [161,162,163]. As for human studies, a promising future direction would be to combine functional neuroimaging with receptor map overlays and gene expression analysis, which would be a non-invasive approach to investigate neuromodulatory mechanisms of time perception and how they are mediated by the activity and connectivity of brain structures.

Another future direction on the level of knowledge systematization could be conducting a meta-analysis on the role of the serotonergic system in time perception. While it is theoretically possible, several challenges currently limit its feasibility. The field suffers from significant uncontrolled variability, such as differences in experimental designs, methodologies, and subject populations across studies. Moreover, the scarcity of studies specifically focusing on the serotonergic system’s role in time perception limits the available data for robust statistical pooling. A meta-analysis requires a sufficient number of high-quality and comparable studies to produce meaningful results. Addressing these limitations would involve standardizing methodologies and increasing the volume of targeted research.

## 5. Conclusions

Summarizing the reviewed literature, we revealed consistent evidence for the role of serotonin (5-HT) in timing behavior but also showed its complex pattern of involvement in different experimental paradigms. While we reconcile the obtained results linking increased serotonergic activation with increased internal time speed, operationalized within the dual klepsydra model as the increased speed of temporal accumulator discharge, there are several research findings that are not in line with this interpretation. Further, our review suggests a link between impulsivity, which we associate with a decelerated internal time speed, and a decrease in serotonergic functioning. However, prospective timing tasks showed greater variability, highlighting the need for further research into how serotonin modulates reward-based temporal decisions with novel approaches to more reliably disentangle the parameters of internal time speed, response inhibition, and other contributing factors.

## Figures and Tables

**Figure 1 ijms-25-13305-f001:**
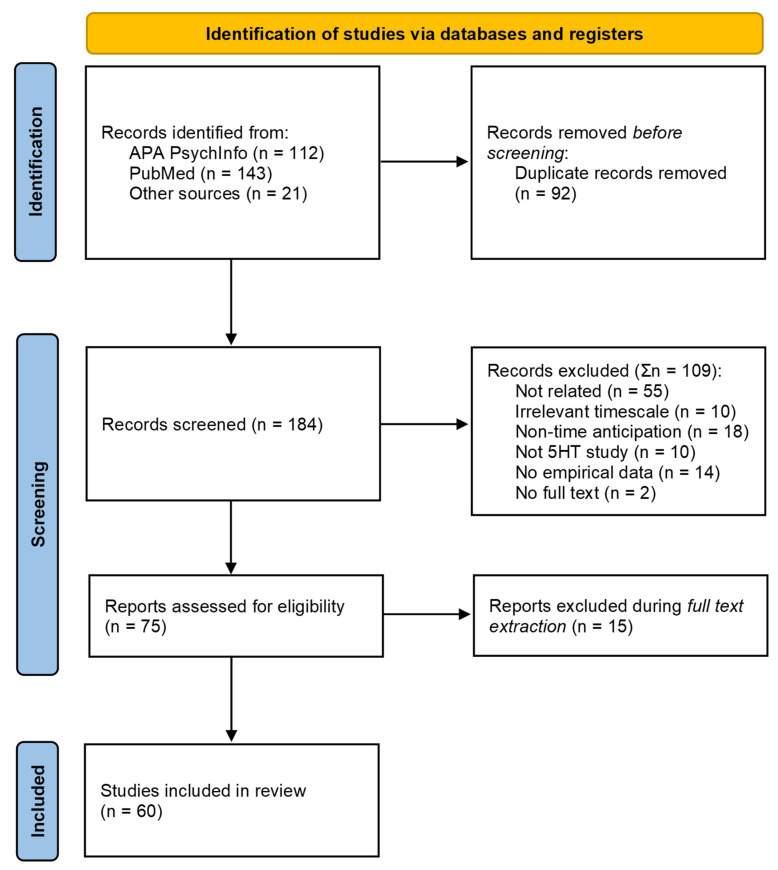
PRISMA flowchart of the study selection process.

**Table 3 ijms-25-13305-t003:** Articles with analysis of serotonergic modulation of behavior in the interval bisection task (IBT).

Article	Sample	Time Perception Measure Protocol and Time Range	Serotonergic Target/ Manipulation	Main Results	Quality Assessment: Total Score
Morrissey et al., 1993 [56]	♀ Wistar rats (n = 30)	IBT 2 s and 8 s	Median and dorsal raphe nuclei lesions with 5,7-DHT	Lesioned group: ↓ D50 WF: n/s	17
Ho et al., 1995 [57]	♀ Wistar rats (n = 48)	IBT 2 s and 8 s +poke requirement modification	Median and dorsal raphe nuclei lesions with 5,7-DHT	Lesioned group: ↓ D50 (in both conditions)	18
Ho et al., 1996 [100]	♀ Wistar rats (n = 22)	IBT 2 s and 8 s	Fluvoxamine acute treatment (15 times for each dose, 4 and 8 mg/kg, i.p.) Desipramine acute treatment (15 times for each dose, 4 and 8 mg/kg, i.p.) Fluvoxamine chronic treatment (8 mg/kg i.p. twice daily for 29 days) Desipramine chronic treatment (8 mg/kg i.p. twice daily for 29 days)	Acute fluvoxamine treatment:↓ WF Acute desipramine treatment:n/s Chronic fluvoxamine and desipramine treatment: n/s for any indices	16
Al-Zahrani et al., 1997 [58]	♀ Wistar rats (n = 28)	IBT 2 s and 8 s +delayed choice modification (8 and 12 s)	Median and dorsal raphe nuclei lesions with 5,7-DHT	↑ D50 in delayed version compared to original ↑ WF in delayed version compared to original D50 increased significantly more pronounced with 12 s delays in the lesioned group compared to the control group	17
Bizot et al., 1997 [72]	♂ Wistar rats (n = 54)	IBT 5 s and 20 s + test 12 s	Clomipramine (4, 8, 16 mg/kg, i.p.) Desipramine (1, 2, 4 and 8 mg/kg, i.p.) Buspirone (0.5, 1, 2 mg/kg, i.p.) Mianserine (0.5, 1, 2, 4 mg/kg, i.p.) d-Amphetamine (0.25, 0.5, 1, 2 mg/kg, i.p.) Haloperidol (0.06, 0.12, 0.5 mg/kg, i.p.) Diazepam (1, 2 and 4 mg/kg, i.p.)	Clomipramine 16 mg/kg: ↑ Omissions, ↑ %L5-CS20; Desipramine 2, 4, 8 mg/kg: ↑ Omissions (only 4 and 8 mg/kg), ↑ %L5-CS12, ↑ %L5-CS20; Buspirone: n/s; Mianserine 2 mg/kg: ↓ %L5-CS5 d-Amphetamine 2 mg/kg: ↑ Omissions, ↓ %L5-CS5, ↓ %L5-CS12 Haloperidol 0.12 mg/kg: ↑ Omissions, ↑ %L5-CS20 Diazepam 4 mg/g: ↑ Omissions, ↑ %L5-CS12, ↑ %L5-CS20	17
T.-J. Chiang et al., 2000 [62]	♀ Wistar rats (n = 23)	IBT 2 s and 8 s	8-OH-DPAT (25, 50, 100, 200 mg/kg, s.c.)	8-OH-DPAT did not alter D50, but dose-dependently ↑ WF	16
Body et al., 2002 [68]	♀ Wistar rats (n = 30)	DTPP: 2.5, 7.5, 12.5, 17.5, 22.5—lever A 27.5, 32.5, 37.5, 42.5, 47.5 s—lever B	Median and dorsal raphe nuclei lesions with 5,7-DHT 8-OH-DPAT (25, 50, 100, 200 mg/kg, s.c.)	Under the vehicle, treatment performance did not differ between lesioned and intact groups 8-OH-DPAT did not alter D50, but dose-dependently ↑ WF in both groups	16
Asgari et al., 2005 [87]	♀Wistar rats (n = 24)	DTPP: 2.5, 7.5, 12.5, 17.5, 22.5—lever A 27.5, 32.5, 37.5, 42.5, 47.5 s—lever B	Quipazine (0.5, 1, 2 mg/kg, s.c.) m-CPBG (2.5, 5, 10 mg/kg, i.p.) Ketanserin (0.5, 1, 2 mg/kg, s.c.) MDL-72222 (0.25, 0.5, 1 mg/kg, i.p.)	Quipazine: ↑ D50 and ↑ WF m-CPBG: n/s Ketanserin reversed the effects of quipazine MDL-72222 did not reverse the effects of quipazine	18
Asgari et al., 2006 [73]	♀ Wistar rats (n = 20)	DTPP: 2.5, 7.5, 12.5, 17.5, 22.5—lever A 27.5, 32.5, 37.5, 42.5, 47.5 s—lever B	DOI (0.0625, 0.125 and 0.25 mg/kg, s.c.) Ketanserin (2 mg/kg, s.c.) MDL-100907 (0.5 and 1 mg/kg, i.p.)	DOI: ↑ D50 and ↑ WF Ketanserin and MDL-100907 did not alter any parameters, but completely blocked effects of DOI, recuperating the initial parameters.	18
Hampson et al., 2010 [74]	♀ Wistar rats (n = 24)	DTPP: 2.5, 7.5, 12.5, 17.5, 22.5—lever A 27.5, 32.5, 37.5, 42.5, 47.5 s—lever B +light intensity modification	DOI (0.0625, 0.125 and 0.25 mg/kg, s.c.) d-Amphetamine (0.2, 0.4 and 0.8 mg/kg, i.p.)	DOI: ↑ D50 for temporal discrimination only D-Amphetamine: ↑ D50 and ↑ WF for temporal and light intensity discrimination;	17
Avlar et al., 2015 [93]	♀ C57BL/6J mice D2R-OE (n = 25) and control (n = 26)	IBT, 6 s and 24 s tone presentations	SB-242084 (0.64–1.0 mg/kg, i.p.)	↓ slope and impaired timing precision in D2R-OE SB-242084: ↓ D50, ↑ slope	16
Halberstadt et al., 2016 [75]	♂ C57BL/6J mice (n = 10)	DTPP. 2.5, 5.0 s—lever A 8.0, 10.5 s—lever B	DOI (0.3, 1.0, 3.0 mg/kg, i.p.) 5CN-NBOH (1, 6 mg/kg, s.c.) MDL-100907 (0.03, 0.1, 0.3 mg/kg, s.c.) SB-242084 (0.1, 0.3, 1 mg/kg, s.c.)	DOI (3 mg/kg), 5CN-NBOH (1 mg/kg): ↑ D50, ↓WF, ↓ slope MDL-100907 (0.3 mg/kg): ↑ D50 SB-242084 (1 mg/kg): ↓ D50 MDL-100907 attenuated DOI effects, but SB-242084 did not	18
Popik et al., 2022 [83]	♂ Sprague-Dawley rats (n = 20)	IBT, 3 s and 12 s light presentations	Psilocybin (0.3 and 1 mg/kg, i.p.) Psilocin (0.22 and 0.72 mg/kg, i.p.) Norpsilocin (0.22 and 0.72 mg/kg, i.p.) (R)-ketamine (7.5, 15, and 30 mg/kg, i.p.) (S)-ketamine (3.75, 7.5, and 15 mg/kg, i.p.)	Psilocybin, Psilocin, and Norpsilocin: n/s (R)-ketamine: n/s (S)-ketamine: ↑ D50 (dose-dependant), ↑ WF	19

Notes: ↑—increase of the parameter; ↓—decrease of the parameter; n/s—non-significant; i.p.—intraperitoneal injection; s.c.—subcutaneous injection; D2R-OE—genetically modified strain with selective and reversible overexpression of D2 receptors in postsynaptic medium spiny neurons in the striatum. %L5-CS5, %L5-CS12, %L5-CS20—percentage of presses on the lever reinforcing signals of 6–10 s duration (L5) after presentation of the conditioned stimuli lasting for 5 s (CS5), 12 s (CS12), and 20 s (CS20), respectively.

**Table 4 ijms-25-13305-t004:** Articles with analysis of serotonergic modulation of behavior in the free-operant psychophysical procedure (FOPP).

Article	Sample	Time Perception Measure Protocol and Time Range	Serotonergic Target/ Manipulation	Main Results	Quality Assessment: Total Score
AI-Zahrani et al., 1996 [59]	♀ Wistar rats (n = 36)	FOPP: 0–25 s—lever A, 25–50 s—lever B	Median and dorsal raphe nuclei lesions with 5,7-DHT	Lesioned group: T50 and ε: n/s ↑ Switching Rate	16
Chiang et al., 1999 [63]	♀ Wistar rats (n = 30)	FOPP: 0–25 s—lever A, 25–50 s—lever B	Median and dorsal raphe nuclei lesions with 5,7-DHT	Lesioned group: T50 and ε: n/s ↑ Switching Rate	19
Chiang et al., 2000 [62]	♀ Wistar rats (n = 23)	FOPP: 0–25 s—lever A, 25–50 s—lever B “Constrained” and “Unconstrained” versions	8-OH-DPAT (25, 50, 100, and 200 mcg/kg s.c.)	“Constrained”version: ↓ T50 (100 and 200 μg/kg) “Unconstrained”version: ↓ T50 (50, 100, and 200 μg/kg) ↑ Switching Rate (200 μg/kg)	16
Body et al., 2001 [64]	♀ Wistar rats (n = 24)	FOPP: 0–25 s—lever A, 25–50 s—lever B	Median and dorsal raphe nuclei lesions with 5,7-DHT; 8-OH-DPAT (100 and 200 mcg/kg, s.c.)	Both lesioned and intact groups: 8-OH-DPAT: ↓T50	18
Body et al., 2002 [55]	♀ Wistar rats (n = 27)	FOPP: 0–25 s—lever A, 25–50 s—lever B	Median and dorsal raphe nuclei lesions with 5,7-DHT; 8-OH-DPAT (100 and 200 mcg/kg, s.c.) WAY-10063 (30, 100, and 300 mcg/kg, s.c.)	Both lesioned and intact groups:8-OH-DPAT: ↓ T50, ↓ ORR WAY-10063: T50: n/s 8-OH-DPAT+WAY-10063: T50 and ORR restored to the baseline	15
Body et al., 2003 [61]	♀ Wistar rats (n = 12)	FOPP: 0–25 s—lever A, 25–50 s—lever B	DOI (0.0625, 0.125 and 0.25 mg/kg, s.c.) Ketanserin (1 mg/kg)	DOI: ↓ T50, ↓ S50 Effects of DOI were blocked by ketanserin	18
Body et al., 2005 [88]	♀ Wistar rats (n = 24)	FOPP: 0–25 s—lever A, 25–50 s—lever B	Quipazine (0.5, 1, and 2 mg/kg, s.c.) m-CPBG (2.5, 5, and 10 mg/kg, i.p.) Ketanserin (0.5, 1, and 2 mg/kg, s.c.) MDL-72222 (0.25, 0.5, and 1 mg/kg, i.p.)	Quipazine: ↓ T50, ↓ S50 and ↓ ORR m-CPBG: n/s Ketanserin reversed the effects of quipazine MDL-72222 did not reverse the effects of quipazine	18
Body et al., 2006a [78]	♀ Wistar rats (n = 20) ♀ Wistar rats (n = 18) ^1^	FOPP: 0–25 s—lever A, 25–50 s—lever B	DOI (0.25 mg/kg, s.c.; and 1 or 3 μg, into dorsal striatum) MDL-100907 (0.5 mg/kg, i.p. or 0.3 μg, into dorsal striatum) RS-102221 (0.15 μg, into dorsal striatum)	DOI: ↓ T50, ↓ S50, ↓ ORR MDL-100907: n/s DOI + MDL-100907: T50, S50, and ORR restored to the baseline n/s effects of intra-striatal DOI, MDL, and RS administration	17
Body et al., 2006b [76]	♀ Wistar rats (n = 14) ♀ Wistar rats (n = 15) ♀ Wistar rats (n = 14) ♀ Wistar rats (n = 14)	FOPP: 0–25 s—lever A, 25–50 s—lever B	DOI (0.25 mg/kg, i.c.) MDL-100907 (0.5 mg/kg, i.p.) SKF-83566 (0.03 mg/kg, i.c.) d-amphetamine (0.4 mg/kg, i.p) Haloperidol (0.05 and 0.1 mg/kg, i.p.)	1st group: ↓ T50: (d-Amphetamine) Haloperidol did not reverse effect of d-amphetamine 2nd group: ↓ T50: (d-Amphetamine) MDL-100907 and SKF-83566 reversed effects of d-amphetamine 3rd group: DOI: ↓ T50 Haloperidol did not reverse effect of DOI 4th group: DOI: ↓ T50 MDL-100907 reversed DOI effects, but SKF-83566 did not	16
Cheung et al., 2007 [77]	♀ Wistar rats (n = 9)	FOPP: 0–40 s—lever A, 40–80 s—lever B	DOI (0.25 mg/kg, i.c.; chronic treatment up to 15 days) d-amphetamine (0.4 mg/kg, i.p.) (−)-quinpirole (0.08 mg/kg, i.p.)	The development of tolerance to DOI’s effect on ↓ T50, but only when animals underwent behavioral training Cross-tolerance to DOI and d-amphetamine (−)-quinpirole: ↓ T50 in animals with tolerance to DOI	16
Body et al., 2009 [60]	♀ Wistar rats (n = 24)	FOPP: 0–25 s—lever A, 25–50 s—lever B	Median and dorsal raphe nuclei lesions with 5,7-DHT (−)-quinpirole (0.08 mg/kg, i.p.) d-amphetamine (0.4 mg/kg, i.p.) SKF-81297 (0.08 mg/kg, s.c.)	d-Amphetamine: ↓ T50 in the sham-lesioned group, but n/s effects were in the 5,7-DHT-lesioned group In both groups: SKF-81297, (−)-quinpirole: ↓ T50	16
Body et al., 2014 [89]	♀ Wistar rats (n = 11–12/group)	FOPP: 0–25 s—lever A, 25–50 s—lever B	Ro-600175 (1, 2, and 4 mg/kg, i.p.) mCPP (0.625, 1.25, and 2.5 mg/kg, i.p.) mCPP+WAY-100635 (0.1 mg/kg, s.c.) mCPP+Isamoltane (0.8 mg/kg, i.p.) mCPP+ MDL-100907 (1 mg/kg, i.p.) mCPP+ SB-242084 (0.6 mg/kg, i.p.)	mCPP (2.5 mg/kg): ↓ T50, ↓ S50, and ↓ ORR Ro-600175 (2 and 4 mg/kg): ↓ ORR SB-242084 + mCPP: ↓↓ T50, ↓↓ S50, and restored ORR Isamoltane, WAY-100635 and MDL-100907: ns Isamoltane has not influenced the effect of mCPP WAY-100635 and MDL-100907 attenuated the effect of mCPP	16

^1^ Half of the group (n = 9) were injected (intra-striatal) with DOI in two doses, and the second half (n = 9) were consequently injected (intra-striatal) with MDL and RS. Notes: ↑—increase of the parameter; ↓—decrease of the parameter; ↓↓—additional decrease of the parameter; n/s—non-significant; i.p.—intraperitoneal injection; s.c.—subcutaneous injection; ORR—overall response rate.

**Table 5 ijms-25-13305-t005:** Articles with analysis of serotonergic modulation of behavior in the fixed-interval peak procedure (PI).

Article	Sample	Time Perception Measure Protocol and Time Range	Serotonergic Target/ Manipulation	Main Results	Quality Assessment: Total Score
Morrissey et al., 1994 [65]	♀ Wistar rats (n = 24)	Fixed Interval 40 s 50% Peak trials	Median and dorsal raphe nuclei lesions with 5,7-DHT	Lesioned group: PRR: n/s T-peak: n/s ↑ ST, ↑ WF	16
Ho et al., 1996 [100]	♀ Wistar rats (n = 22) ♀ Wistar rats (n = 36)	Fixed Interval 30 s 50% Peak trials	Fluvoxamine acute treatment (15 times for each dose, 4 and 8 mg/kg, i.p.) Desipramine acute treatment (15 times for each dose, 4 and 8 mg/kg, i.p.) Fluvoxamine chronic treatment (8 mg/kg i.p. twice daily for 29 days) Desipramine chronic treatment (8 mg/kg i.p. twice daily for 29 days)	Acute fluvoxamine (8 mg/kg i.p.): ↑ PRR Acute desipramine (8 mg/kg i.p.):↓ PRR ↓ PRR Chronic fluvoxamine treatment: ↓ T-peak Chronic desipramine treatment: n/s	16
Bayley et al., 1998 [102]	♂ PVG Black hooded rats (n = 60)	Fixed Interval 72 s 40% Peak trials	Antidepressants (SERT as a key target): Imipramine (1, 3, or 10 mg/kg, i.p.) Zimelidine (10, 20, or 40 mg/kg, i.p.) Clomipramine (1, 3, or 10 mg/kg, i.p.) Other agents: d-Amphetamine (0.5, 0.75, or 1.5 mg/kg, s.c.) Diazepam (1, 3, or 5 mg/kg, i.p.)	Zimelidine, 40 mg/kg: ↓ T peak, ↓ ST, and ↓ Mean lever presses/min d-amphetamine: ↓ Mean lever presses/min and ↑ ST Other drugs: ↓ Mean lever presses/min	13
Asgari et al., 2006 [69]	♀ Wistar rats (n = 30)	Fixed Interval 30 s 50% peak trials	8-OH-DPAT (0.05 mg/kg, s.c.) WAY-100635 (0.1 mg/kg, s.c.) DOI (0.25 mg/kg, s.c.) Ketanserin (0.2 mg/kg, s.c.)	8-OH-DPAT: ↓ PRR, ↓ T-peak, and ↑ WF DOI: ↓ T-peak and ↑ WF WAY-100635 blocked effects of 8-OH-DPAT on T-peak, but PRR was still reduced Ketanserin blocked effects of DOI	16
MacDonald and Meck 2005 [95]	♂ Sprague-Dawley rats (n = 20)	Fixed interval Peak procedure 10, 30, and 90s with separate levers for each duration (total of 3)	Clozapine (0.6, 1.2, and 2.4 mg/kg, i.p.) Haloperidol (0.03, 0.06, and 0.12 mg/kg, i.p.)	Clozapine: ↓ T-peak Haloperidol: ↑ T-peak	13
Buhusi and Meck 2007 [96]	♂ Sprague-Dawley rats (n = 10)	Fixed Interval 20 s 50% Peak trials With gaps during the trial and episodically turned off light	Clozapine (2 mg/kg, i.p.)	Clozapine in PI: ↓ T-peak, ↓ PRR Clozapine in gap condition: ↓ T-peak	13
Heilbronner and Meck 2014 [97]	♂ Sprague-Dawley rats (n = 23)	Fixed interval of 40 s with 50% peak trials bi-fixed intervals peak procedure 10 and 40 s (rewarded by 4 pellets) intermixed with defection trails of small immediate reward (1 pellet). 3 levers: 10 s, 40 s, and defection.	Fluoxetine (3, 5, or 8 mg/kg, i.p.) Cocaine (15 mg/kg, i.p.) Methamphetamine (1 mg/kg, i.p.)	Fluoxetine: ↓ defection response Cocaine, Methamphetamine: ↑ defection response, ↑ ST Negative correlation between T-peak for 10 s and number of defection response n/s T-peak shift for all drug treatments	13
Shapiro et al., 2018 [70]	♂ Sprague-Dawley rats (n = 20) ♂ Sprague-Dawley rats (n = 10)	Bi-fixed intervals peak procedure 5 and 20 s associated with interoceptive states induced by saline and 8-OH-DPAT, 25% of Peak Trials Bi-fixed intervals peak procedure 10 s (tone) and 20 s (light); separate and combined for testing. 50% of Peak Trials	8-OH-DPAT (0.25 mg/kg, i.p.) WAY-100635 (0.1 mg/kg, i.p.)	T-peak was centered around 7 s under 8-OH-DPAT administration regardless of reinforced interval (5 or 20 s) WAY-100635 had no effect on timing when tone and light were presented separately, but disrupted timing when combined	13
Dellu-Hagedorn et al., 2018 [15]	♂ Wistar Han rats (n = 35)	Fixed Interval Schedule 60 s	High- vs. Low-impulsivity group (based on total response) HPLC: 5-HIAA, 5-HT, and DA, DOPAC, and DOPAC/DA concentrations	High- vs. Low-impulsivity group: turnover rates of 5-HT ↑ in the prelimbic and posterior insular cortex, Hippocampus (HPC), and Ventrolateral Striatum (VLS); ↓ in the dorsolateral orbitofrontal cortex; turnover rates of DA ↑ in the prelimbic cortex and VLS Positive correlations between total activity and serotonergic turnover in the HPC and VLS	17
Acosta et al., 2018 [123]	♂ VPA-SAL (n = 5) * ♂ VPA-VPA (n = 5) ♀ SAL-SAL (n = 6) ♀ VPA-VPA (n = 6)	Fixed 15 and 45 s Intervals with 50% of Peak trials	Prenatal exposure to VPA—a treatment used to induce human-like autistic features in rodents HPLC: 5-HT, 5-HIAA, and DA and DOPAC concentrations in the dorsal striatum;	↓ T-peak and ↓ PRR in the VPA-VPA group Both female and male VPA mice present decreased timing accuracy and precision—↑ ST Early social enrichment reversed timing deficits in male VPA mice n/s differences for 5-HT and 5-HIAA levels between groups ↑ DA and ↓ DA turnover level in VPA-VPA female mice	18
Miyazaki et al., 2020 [16]	♂ C57BL/6J bigenic (n = 27) and wild-type (n = 15) mice	Fixed intervals of 6 s and 10 s (certain condition) and 4, 6, and 8 s and 2, 6, and 10 s (uncertain condition) with a 75% reward probability in a sequential tone–food waiting task	Optogenetic activation of serotonergic neurons via optic fibers implanted in both the mPFC and the DRN	↑T-peak under stimulation in the OFC and in the DRN Stimulation in the NAc: n/s ↑ T-peak under serotonin stimulation in the mPFC only in the uncertain condition	20

* HPLC—High-performance liquid chromatography; VPA—valproic acid. VPA-SAL—Prenatal VPA treatment with further growing environment with control mice. VPA-VPA—Prenatal VPA treatment with further growing environment with VPA prenatally treated mice. SAL-SAL—Prenatal saline treatment with further growing environment with control mice. Notes: ↑—increase of the parameter; ↓—decrease of the parameter; n/s—non-significant; i.p.—intraperitoneal injection; s.c.—subcutaneous injection; HPLC—High-performance liquid chromatography.

**Table 6 ijms-25-13305-t006:** Articles with analysis of the serotonergic modulation of behavior in differential reinforcement of low rate (DRL) paradigm.

Article	Sample	IRT Range	Serotonergic Target/ Manipulation	Main Results	Quality Assessment: Total Score
Wogar et al., 1992 [66]	♀ Wistar rats (n = 35)	>15 s	Median and dorsal raphe nuclei lesions with 5,7-DHT	Lesioned group: ↓ PkL ↑ PkA ↓ Reinforcement rates	17
Wogar et al., 1993 [67]	♀ Wistar rats (n = 68)	>15 s	Median and dorsal raphe nuclei lesions with 5,7-DHT	Lesioned group: ↓ Reinforcement rates, ↓ PkL ↑ PkA. n/s response rates Magnitude of reinforcement (sucrose concentration) did not alter any performance parameters 80% of body weight group compared with 90% of body weight animals: ↓ mean response rate, ↓ PkA and ↑ PkL	17
Sokolowski and Seiden 1999 [99]	♂ Sprague-Dawley rats (n = 22)	>72 s	Fluoxetine Sertraline Paroxetine All drugs were injected i.p. at doses of 5, 10, and 20 mg/kg	↑ Reinforcement rate (SSRIs) ↓ Response rate (sertraline and fluoxetine) ↑ PkL (sertraline and paroxetine) ↓ PkA (paroxetine)	17
Sabol et al., 2000 [103]	♂ Sprague-Dawley rats (n = 72)	>72 s	Methamphetamine (15 mg/kg injections) *	n/s PkL ↓ PkA 5HT decreased in Frontal, Somatosensory, and Occipital Cortices, Striatum, Amygdala, Hipp, and NAc	14
Scott-McKean et al., 2008 [91]	♂ C57BL/6J mice wild-type (n = 12) and 5HTR1A KO mice (n = 12)	>72 s	Fluoxetine (2.5, 5, 10, and 20 mg/kg, i.p.) Desipramine (1.25, 2.5, 5, and 10 mg/kg, i.p.) WAY-100635 (0.03 mg/kg, i.p.)	5HTR1A KO mice: ↑ Reinforcement rate (Fluoxetine, Desipramine) ↓ Response rate (Fluoxetine, Desipramine) ↑ PkL (Fluoxetine, Fluoxetine+WAY, Desipramine) ↑ PkA (Fluoxetine+WAY, Desipramine) WT mice: ↑ Reinforcement rate (Desipramine) ↓ Response rate (Fluoxetine+WAY, Desipramine) ↑ PkL (Fluoxetine+WAY, Desipramine) ↑ PkA (Fluoxetine+WAY, Desipramine)	18
Malikowska-Racia et al., 2023 [98]	♂ Sprague Dawley rats (n = 32) ♂ Sprague Dawley rats (n = 36)	>72 s	Fluoxetine (2.5, 5, and 10 mg/kg, i.p.) (S)-Ketamine (3.75, 7.5, 15, and 30 mg/kg, i.p.) (R)-Ketamine (3.75, 7.5, 15, and 30 mg/kg, i.p.)	↑ Reinforcement rate (Fluoxetine, (R)- and (S)-Ketamine) ↓ Response rate (Fluoxetine, (S)-Ketamine) ↑ PkL (Fluoxetine, (S)-Ketamine) ↓ PkA ((S)-Ketamine)	17
Malikowska-Racia et al., 2023 [82]	♂ n = 36 Sprague Dawley rats	>72 s	Psilocybin (1 mg/kg, i.p.) LSD (0.08 mg/kg, i.p.)	↑ Reinforcement rate (Psilocybin, LSD) ↑ PkL (Psilocybin)	18

* The route of administration is not specified. Notes: IRT—inter-response time; ↓—decrease of the parameter; ↑—increase of the parameter; n/s—non-significant changes; i.p.—intraperitoneal injection.

**Table 7 ijms-25-13305-t007:** Articles with analysis of serotonergic modulation of behavior in the delay discounting task.

Article	Sample	Time Perception Measure Protocol and Time Range	Serotonergic Target/ Manipulation	Main Results	Quality Assessment: Total Score
Matsunami et al., 2012 [101]	♂ domestic chicks (n = 193)	Delay Discounting SS—0 s, 1 bead; LL—1.5 s, 6 grains	Fluvoxamine (10–20 mg/kg, i.p.)	↑ %LL (Fluvoxamine) SSRI increases 5HT and DA levels in the medial striatum and NAc	14
Zaichenko et al., 2013 [71]	♂ Wistar rats (n = 40)	Delay DiscountingSS—0 s, 1 pellet LL—5 s, 4 pellets	Impulsive vs. Non-impulsive animals 8-OH-DPAT (0.1 mg/kg, i.p.) WAY-100635 (0.2 mg/kg, i.p.)	WAY-100635 ns ↑ K_0_ (8-OH-DPAT) Impulsive animals: ↑ %LL (8-OH-DPAT)	17
Zaichenko et al., 2014 [80]	♂ Wistar rats (n = 38)	Delay Discounting SS—0 s, 1 pellet LL—5 s, 4 pellets	Impulsive vs. Non-impulsive animals DOI (0.2 mg/kg, i.p.) ketanserin (0.2 mg/kg, i.p.)	Both groups: ↑ %LL (DOI) Impulsive group: ↓ K_0_ (DOI) ↓ %LL (ketanserin)	17
Kirkpatrick et al., 2014 [129]	♂ Sprague-Dawley rats Isolated condition (IC; n = 12) Enriched condition (EC; n = 11)	Delay Discounting SS—10 s, 1 pellet LL—30 s, 2–3 pellets	Isolated condition vs. enriched condition Neurochemical analysis: NE, DA, DOPAC, 5-HT, and 5-HIAA concentrations within the prefrontal cortex (PFC) and nucleus accumbens (NAc).	no significant differences in monoamine levels between the groups NAc 5-HIAA and NE concentrations positively correlate with the %LL	13
Buhusi et al., 2017 [90]	♂ CHL1-deficient (KO, n = 16) and wild-type (WT, n = 16) mice	Delay Discounting SS—0 s, 1 pellet LL—0–64 s, 4 pellets	Ro60-0175 (0.6 and 1.2 mg/kg i.p.)	Chronically stressed CHL1-KO: ↓ %LL (chronic stress) ↑ %LL (Ro60-0175) Decreased number of cFos-positive neurons in the prelimbic cortex and dorsal striatum Chronically stressed WT: ↑ %LL (Ro60-0175)	17
Dellu-Hagedorn et al., 2018 [15]	♂ Wistar Han rats (n = 32)	Delay Discounting SS—0 s, 1 pellet LL—0–40 s, 5 pellets	High- vs. Low-discounting rats (based on LL%) Neurochemical analysis: DA, DOPAC, 5-HT, and 5-HIAA concentrations	High- vs. Low-discounting rats: ↓ basal turnover rates of both serotonin and dopamine exhibited in the amygdala central nucleus (CE), ventrolateral striatum (VLS), and nucleus accumbens core	17
Jiang et al., 2022 [79]	♂ Sprague-Dawley rats (n = 32)	Delay Discounting SS—0 s, 1 pellet LL—0–16 s, 5 pellets	DOI (0.5 and 1 mg/kg, i.p.) DOI (1 mg/kg, i.p.) + Ketanserine (1 mg/kg, s.c.) DOI (1 mg/kg, i.p.) + SB-242084 (1 mg/kg, i.p.) Lisuride (0.1, 0.3, and 0.5 mg/kg, i.p.) Lisuride (0.3 mg/kg, i.p.) + ketanserin (1 mg/kg) Lisuride (0.3 mg/kg, i.p.) + WAY-100635 (1 mg/kg, i.p.) Lisuride (0.3 mg/kg, i.p.) + L-745870 (1 mg/kg, i.p.) Lisuride (0.3 mg/kg, i.p.) + Tiapride (40 mg/kg, i.p.).	↓ %LL, ↑ K_0_ (DOI) The effects of DOI were blocked by ketanserin, but not by SB-242084. ↑ %LL, ↑ K_0_ (Lisuride) The effects of lisuride were not antagonized by ketanserin, WAY-100635, or L-745870, but were attenuated by tiapride.	17
Roberts et al., 2023 [84]	♂ Long Evans rats (n = 20) ♀ Long Evans rats (n = 20)	Delay Discounting Task SS—2 s, 1 pellet LL—2–52 s, 3 pellets	Psilocybin (1 mg/kg, gavage)	Psilocybin treatment did not affect decision-making in the delay discounting task in both male and female rats	17

Notes: SS—smaller but sooner reward; LL—larger, later reward; ↓—decrease of the parameter; ↑—increase of the parameter; i.p.—intraperitoneal injection; s.c.—subcutaneous injection; NAc—Nucleus Accumbens; CHL1-deficient mice—Close Homolog of L1 (CHL1) gene knock-out mice.

**Table 8 ijms-25-13305-t008:** Articles with analysis of serotonergic modulation of behavior in other behavioral tests.

Article	Sample	Time Perception Measure Protocol and Time Range	Non-Temporal Measures	Serotonergic Target/ Manipulation	Main Results	Quality Assessment: Total Score
Inaba et al., 2013 [130]	♂ Rhesus monkeys (n = 2)	Go/No-Go Task 1–3 steps with 400–1200 ms anticipation	Electrophysiology records for 98 single neurons in the dorsal raphe of two monkeys	Dorsal raphe nuclei neurons	Neurons in the dorsal raphe nucleus show responses related to schedule onset, reward expectation, reward outcome, and reward amount	16
McLaughlin et al., 2017 [92]	♂ Sprague-Dawley rats (n = 24)	Fixed consecutive number (FCN) task (2–5 s) Variable consecutive number task with discriminative stimulus (VCN-SD)	Three doses of CB1 ligands and effects of combination with WAY treatment	AM 251 (CB1 inverse agonist) AM 6527 (CB1 antagonist) AM 6545 (peripherally restricted CB1 antagonist) WAY-100635 (WAY)	AM 251, AM 6527, and AM 6545 n/s on accuracy or chain lengths reduced accuracy in FCN (WAY + AM 251 and AM 6527). impairment in VCN-SD (WAY + AM 251)	16

Notes: n/s—non-significant.

## Data Availability

Not applicable.

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
