# Peer review of "The Role of the Serotonergic System in Time Perception: A Systematic Review"

_ijms, 2024, doi:10.3390/ijms252413305_

Round 1
Reviewer 1 Report
Comments and Suggestions for Authors
This review deals with the transmitter and neuromodulator serotonin and the serotonergic system in time perception, collecting data from studies in various species, highlighting serotonin´s complex involvement across retrospective, immediate, and prospective timing paradigms. The review focusses on the mechanisms underlying interval timing and time perception as perception of duration in the scale of several seconds, using a translational approach. The authors claim to analyze evidence obtained in a wide range of timing schedules and paradigms and bind together the facts revealed in human and animal studies. Lastly, the authors propose a concept of the serotonergic system role in time perception.
First of all, the review is convincing simply because of its length of 67 pages dealing with all possible aspects. It is a very diligent work that summarizes the state of science on this topic.
However, I am a bit astonished that the first and second authors of this review cannot be found in PubMed. Thus, unlike authors 3-5, they do not appear to be experienced scientists.
Generally, I strongly recommend to update the review, as it now ends May 2023.
A bit time consuming is, that the references listed in the Tables are not signed by the numbers listed in the reference list – that must be supplemented.
As far as I can see (and I am overwhelmed by the flood of data), the review seemingly is written diligently and correctly. I must say I am not able to control each sentence and data mentioned.
Author Response
<<Thank you very much for taking the time to review this manuscript. Please find the detailed responses below and the corresponding revisions/corrections highlighted in the re-submitted files. Below you can see point-by-point responses to your comments.
This review deals with the transmitter and neuromodulator serotonin and the serotonergic system in time perception, collecting data from studies in various species, highlighting serotonin´s complex involvement across retrospective, immediate, and prospective timing paradigms. The review focuses on the mechanisms underlying interval timing and time perception as perception of duration in the scale of several seconds, using a translational approach. The authors claim to analyze evidence obtained in a wide range of timing schedules and paradigms and bind together the facts revealed in human and animal studies. Lastly, the authors propose a concept of the serotonergic system role in time perception.
First of all, the review is convincing simply because of its length of 67 pages dealing with all possible aspects. It is a very diligent work that summarizes the state of science on this topic.
Comment 1. However, I am a bit astonished that the first and second authors of this review cannot be found in PubMed. Thus, unlike authors 3-5, they do not appear to be experienced scientists.
<<Response1: Indeed, first authors are PhD students, but their work was tentatively guided and supervised; thus, all the authors equally significantly contributed to the manuscript. Also, the second author's surname was misspelled (correct Mitiureva) and her works can be found in PubMed.
Comment 2: Generally, I strongly recommend to update the review, as it now ends May 2023.
<< Response 2: We have addressed your concern and refreshed our search for the years 2023 and 2024 and found 7 articles (DOIs: 10.1038/s41380-024-02696-1, 10.1523/JNEUROSCI.0602-24.2024, 10.3389/fnbeh.2023.1289520, 10.1021/acsmedchemlett.3c00406, 10.1016/j.jpsychires.2023.08.005, 10.1016/j.cois.2023.101034, 10.1017/S0033291721005493).
Among them, only one article meets our inclusion criteria, and it was already included in our systematic review (Medvedeva et al., 2023). Thus, all the relevant existing studies are included. We have further specified in the 2.1. Literature Search Strategy section that we have refreshed the search: “The database search was conducted in May 2023 (and refreshed in December 2024) with no restrictions to the year of publication.”
Comment 3: A bit time consuming is that the references listed in the Tables are not signed by the numbers listed in the reference list – that must be supplemented.
<< Response 3: We have added number-like citations to the tables.
Comment 4: As far as I can see (and I am overwhelmed by the flood of data), the review seemingly is written diligently and correctly. I must say I am not able to control each sentence and data mentioned.
<< Response 4: Thank you for the review and comments.
Reviewer 2 Report
Comments and Suggestions for Authors
This is a very long (60 pages) review article on how manipulations of the 5HT (serotonergic) neural system in humans or experimental animals affects performance on several tasks related to perception of time.
Ever since Albert Einstein presented the concept of "space-time" in his theory of General Relativity in the early 20th century, many physicists, philosophers and mathematicians have presented their views on what time actually is. One of the most revealing conclusions I have read is that "time is what clocks measure". In that context, the authors of this complex review article present how the neuroscience/neuropharmacology community has attempted to characterize how humans and experimental animals measure past, current and future time.
Their task is complicated by several factors. First, the 5HT neural system projects widely to many forebrain areas. Second, there are many different 5HT receptors, mostly post-synaptic but also some important presynaptic, that potentially modulate neuronal firing in opposite directions in these various forebrain areas. Third, experiments on humans are limited by ethical considerations that do not apply to experimental animals. For instance, as the authors present, many experiments in animals involve lesions of or ontogenetic/pharmacological manipulations (like direct brain lesions or injections) of the 5HT neural system that are not possible in humans. Humans, on the other hand, can communicate verbally to the experimenters. Animals (at the moment) cannot do this, so experimenters asses behavior and first must "train" the animals to behave. Fourth, one of the commonly used methods for increasing brain 5HT activity is oral administration of the amino acid and 5HT precursor L-tryptophan. We now know that 90-95% of oral L-tryptophan is not converted to 5HT, but rather is metabolized to kynurenine and subsequently to kynurenine metabolites by the intestinal microbiome. These Kyn metabolites are absorbed into the bloodstream and have a variety of effects on multiple CNS neurotransmitter systems. So, while L-tryptophan administration certainly raises brain 5HT levels, it also has more widespread effects. Fifth, most drugs used to stimulate or block 5HT receptors are "dirty", meaning they are promiscuous and can act at other receptors. both 5HT and other neurotransmitters. So, interpreting their effects in terms of specific 5HT actions is limited. (There may be other limitations I haven't thought of)
I spite of these limitations, the authors have provided an extensive compendium of research for the community. This is, thus, a very valuable review article and represents a tremendous effort of the part of the authors, which they present in a nicely structured format with several summary Tables.
Obviously, time perception involves the cognitive function of memory, and any study of time perception necessarily involves memory function. The authors restrict their review to studies that involve the time range of seconds to minutes, and exclude studies of hours, days or longer. Of course, these longer intervals are also of interest, but I support the authors' restriction of articles reviewed.
Any study of this nature will necessarily be subject to the criticism of bias in selection of papers to summarize. The authors discuss this problem and justify their approach.
Admittedly, it is difficult to draw any conclusions in this complex area, and the authors are appropriately circumspect about what the literature is telling all of us. One very general conclusion is that activation of the 5HT system accelerates time perception and reduction of the the 5HT system activity reduces time perception. But this is perhaps too broad and has too many exceptions.
I reviewed their Supplementary Tables and Figure and feel that all of these are relevant and should be included.
There are a very few minor English errors which can be easily corrected during final editing.
Author Response
<<Thank you very much for taking the time to review this manuscript. Please find the detailed responses below and the corresponding revisions/corrections highlighted in the re-submitted files. Below you can see point-by-point responses to your comments.
This is a very long (60 pages) review article on how manipulations of the 5HT (serotonergic) neural system in humans or experimental animals affects performance on several tasks related to perception of time.
Ever since Albert Einstein presented the concept of "space-time" in his theory of General Relativity in the early 20th century, many physicists, philosophers and mathematicians have presented their views on what time actually is. One of the most revealing conclusions I have read is that "time is what clocks measure". In that context, the authors of this complex review article present how the neuroscience/neuropharmacology community has attempted to characterize how humans and experimental animals measure past, current and future time.
Comment 1: Their task is complicated by several factors. First, the 5HT neural system projects widely to many forebrain areas. Second, there are many different 5HT receptors, mostly post-synaptic but also some important presynaptic, that potentially modulate neuronal firing in opposite directions in these various forebrain areas. Third, experiments on humans are limited by ethical considerations that do not apply to experimental animals. For instance, as the authors present, many experiments in animals involve lesions of or ontogenetic/pharmacological manipulations (like direct brain lesions or injections) of the 5HT neural system that are not possible in humans. Humans, on the other hand, can communicate verbally to the experimenters. Animals (at the moment) cannot do this, so experimenters assess behavior and first must "train" the animals to behave. Fourth, one of the commonly used methods for increasing brain 5HT activity is oral administration of the amino acid and 5HT precursor L-tryptophan. We now know that 90-95% of oral L-tryptophan is not converted to 5HT, but rather is metabolized to kynurenine and subsequently to kynurenine metabolites by the intestinal microbiome. These Kyn metabolites are absorbed into the bloodstream and have a variety of effects on multiple CNS neurotransmitter systems. So, while L-tryptophan administration certainly raises brain 5HT levels, it also has more widespread effects. Fifth, most drugs used to stimulate or block 5HT receptors are "dirty", meaning they are promiscuous and can act on other receptors. both 5HT and other neurotransmitters. So, interpreting their effects in terms of specific 5HT actions is limited. (There may be other limitations I haven't thought of)
<< Response 1: Thank you for pointing this out. While we acknowledge most of these limitations already in the review, we like the way you frame them and dare to include some of your sentences unchanged into our text. We also acknowledge the limitations you have mentioned. In human studies, those with L-tryptophan administration were ranked lower in a corresponding question of quality assessment. Although many of your concerns were mentioned in the limitations section, we have addressed additional ones in this paragraph in Limitations:
“Additionally, in the current review, we provide integration of the results within the DKM framework. Although it seems to reconcile many different studies, there are certainly many unresolved questions and unproved assumptions that should be studied further. While we managed to draw a general conclusion, there are discrepancies on the scale of particular receptor types and structures implicated and how their involvement influences time perception. Our task was complicated by several factors. First, the 5HT neural system projects widely to many forebrain areas. Second, there are many different 5HT receptors, mostly post-synaptic but also some important presynaptic, that potentially modulate neuronal firing in opposite directions in these various forebrain areas”
“Additionally, in the current review, we provide integration of the results within the DKM framework. Although it seems to reconcile many different studies, there are certainly many unresolved questions and unproved assumptions that should be studied further. While we managed to draw a general conclusion, there are discrepancies on the scale of particular receptor types and structures implicated and how their involvement influences time perception. The results of our review regarding differential involvement of particular serotonergic receptors to the interval timing in particular paradigm should be considered with caution as this might be due to the uneven distribution of studies targeting different serotonergic receptors among paradigms as could be seen in Table 1. It is also important to mention that serotonergic neurotransmission is tightly connected with other neurotransmitter systems, which may induce a chain of mutual influences and mixed effects on timing behavior [173–175]. For instance, dopaminergic neurotransmission also plays a crucial role in temporal cognition, which is discussed in a number of comprehensive reviews [23,176,177]. Moreover, many substances used to manipulate serotonergic neurotransmission often induce widespread and mixed effects involving different serotonin receptor types and even those of other neurotransmitters.”
In spite of these limitations, the authors have provided an extensive compendium of research for the community. This is, thus, a very valuable review article and represents a tremendous effort of the part of the authors, which they present in a nicely structured format with several summary Tables.
Obviously, time perception involves the cognitive function of memory, and any study of time perception necessarily involves memory function. The authors restrict their review to studies that involve the time range of seconds to minutes, and exclude studies of hours, days or longer. Of course, these longer intervals are also of interest, but I support the authors' restriction of articles reviewed.
Any study of this nature will necessarily be subject to the criticism of bias in selection of papers to summarize. The authors discuss this problem and justify their approach.
Comment 2: Admittedly, it is difficult to draw any conclusions in this complex area, and the authors are appropriately circumspect about what the literature is telling all of us. One very general conclusion is that activation of the 5HT system accelerates time perception and reduction of the 5HT system activity reduces time perception. But this is perhaps too broad and has too many exceptions.
<< Response 2: Thank you for your thoughtful consideration, yet we provided such a general interpretation with high cautious bearing in mind its limitations, which were discussed in a corresponding section.
I reviewed their Supplementary Tables and Figure and feel that all of these are relevant and should be included.
There are a very few minor English errors which can be easily corrected during final editing.
<< Thank you for the review. We corrected the text accordingly.
Reviewer 3 Report
Comments and Suggestions for Authors
“The role of the serotonergic system in time perception: a systematic review”(ijms-3339549)
This manuscript aims to provide the state-of-art of the role of serotonergic system in time perception by a systematic review. Sixty papers were selected for full-text review, encompassing both human (n = 10) and animal studies (n = 50). The results revealed consistent evidence for the role of serotonin in timing behavior, highlighting its complex involvement across retrospective, immediate, and prospective timing paradigms. Increased serotonergic activation aligns with increased internal time speed, interpreted within the dual klepsydra model as accelerated temporal accumulator discharge, though some findings diverge from this view. Additionally, we link impulsivity—associated with decreased serotonergic functioning in our review—to slower internal time speed. Overall, this topic is interesting and important. The findings are relatively comprehensive and important for the development for this field. However, some concerns appeared after reading the whole manuscript.
1. Some previous related reviews can not be neglected and should be reviewed and discussed, such as,
Marinho, V., Oliveira, T., Rocha, K., Ribeiro, J., Magalhães, F., Bento, T., ... & Teixeira, S. (2018). The dopaminergic system dynamic in the time perception: a review of the evidence. International Journal of Neuroscience, 128(3), 262-282.
2. PRISMA should be updated to the latest version.
Page, M. J., McKenzie, J. E., Bossuyt, P. M., Boutron, I., Hoffmann, T. C., Mulrow, C. D., ... & Moher, D. (2021). The PRISMA 2020 statement: an updated guideline for reporting systematic reviews. bmj, 372.
3. Why only included “PubMed, APA PsycINFO and APA PsycARTICLES”? The datasets with more coverage of literature should also be included, such as Web of Science and Scopus.
4. Did you also screen the reference list of the selected paper to find the potential related papers?
5. Is there any possible to provide statistical results, such as meta-analysis for this topic.
6. Since the current literature search was conducted in May 2023, and more than one and half year has passed. It would be better to update the literature search results to reflect the most recent update of this topic.
Author Response
<<Thank you very much for taking the time to review this manuscript. Please find the detailed responses below and the corresponding revisions/corrections highlighted in the re-submitted files. Below you can see point-by-point responses to your comments.
This manuscript aims to provide the state-of-art of the role of serotonergic system in time perception by a systematic review. Sixty papers were selected for full-text review, encompassing both human (n = 10) and animal studies (n = 50). The results revealed consistent evidence for the role of serotonin in timing behavior, highlighting its complex involvement across retrospective, immediate, and prospective timing paradigms. Increased serotonergic activation aligns with increased internal time speed, interpreted within the dual klepsydra model as accelerated temporal accumulator discharge, though some findings diverge from this view. Additionally, we link impulsivity—associated with decreased serotonergic functioning in our review—to slower internal time speed. Overall, this topic is interesting and important. The findings are relatively comprehensive and important for the development of this field. However, some concerns appeared after reading the whole manuscript.
Comment 1. Some previous related reviews can not be neglected and should be reviewed and discussed, such as,
Marinho, V., Oliveira, T., Rocha, K., Ribeiro, J., Magalhães, F., Bento, T., ... & Teixeira, S. (2018). The dopaminergic system dynamic in the time perception: a review of the evidence. International Journal of Neuroscience, 128(3), 262-282.
<< Response 1: Thank you for the point. While our review’s focus was on serotonergic functioning we have also added mentioning of the reviews tackling the role of the dopaminergic system in Discussion: “It is also important to mention that serotonergic neurotransmission is tightly connected with other neurotransmitter systems, which may induce a chain of mutual influences and mixed effects on timing behavior [173–175]. For instance, dopaminergic neurotransmission also plays a crucial role in temporal cognition, which is discussed in a number of comprehensive reviews [23,176,177].” (e.g. Marihno et al., 2018 and Fung et al., 2021.) Such reviews are more common to the field; however, the discussion of interaction between the serotonergic and dopaminergic systems in serving time perception is beyond the scope of the present study.
We also citing other relevant reviews:
“A range of comprehensive reviews on serotonergic modulation of time perception has been published to date. The review provided by Bradshaw and Szabadi group highlighted the role of serotonin in animal timing behavior [14]. Subsequently, timing behavior has been studied in its relation to motivation, and how it is altered in diverse neuropathologies [21,22]. Marinho and colleagues explored genetic bases of time perception both in humans and animals [23]. Agostino and colleagues considered specific neural circuits with proposed neurotransmitter regulation [9]. Additionally, the effects of psychedelic treatments on timing have been discussed [24–26], offering a unique avenue for studying behavioral alterations through interventions with specific substances. However, the evidence towards serotonergic system involvement in time perception has not been systematically reviewed within the translational approach.“
Nonetheless, it is important to note that the majority of them did not incorporate answering to the majority of questions we pose; thus, we do not largely argue or compare our theses with theirs.
Comment 2. PRISMA should be updated to the latest version.
Page, M. J., McKenzie, J. E., Bossuyt, P. M., Boutron, I., Hoffmann, T. C., Mulrow, C. D., ... & Moher, D. (2021). The PRISMA 2020 statement: an updated guideline for reporting systematic reviews. bmj, 372.
<< Response 2: We followed the latest PRIZMA guideline but mis-cited it, and we have addressed this issue by updating the citation.
Comment 3. Why only included “PubMed, APA PsycINFO and APA PsycARTICLES”? The datasets with more coverage of literature should also be included, such as Web of Science and Scopus.
<< Response 3: We would like to use these databases, but we have issues accessing them. Three used databases are generally sufficient for our topic, and we conducted hand search and citation analysis additionally. We have further specified our choice for databases in Methods and discussed potential limitations.
Methods: “PubMed was chosen for its extensive coverage of biomedical literature, including neuroscience and mental health topics, which are central to our research focus. APA PsycINFO and APA PsycARTICLES were selected for their specialization in psychology and behavioral sciences, ensuring the inclusion of high-quality, peer-reviewed articles pertinent to our review. Together, these databases provide a robust foundation for identifying relevant studies across disciplines central to our research question.”
Limitations: “While PubMed, APA PsycINFO, and APA PsycARTICLES provide extensive coverage of biomedical and psychological literature, they may not fully encompass studies from certain interdisciplinary domains. To address this, we supplemented our database searches with hand-searching and citation tracking to identify additional relevant studies. However, including broader interdisciplinary databases, such as Scopus or Web of Science, might have captured additional studies from fields adjacent to our primary focus. Future reviews could consider integrating such databases to further enhance comprehensiveness.”
Comment 4. Did you also screen the reference list of the selected paper to find the potential related papers?
<< Response 4: Yes, it was specified in Methods: “Additionally, further papers (n = 21) were identified in reference lists of retrieved articles and screened for eligibility (backward snowballing).”
Comment 5. Is there any possibility to provide statistical results, such as meta-analysis for this topic.
<< Response 5: While conducting a meta-analysis on the role of the serotonergic system in time perception is theoretically possible, several challenges currently limit its feasibility. The field suffers from significant uncontrolled variability, such as differences in experimental designs, methodologies, and subject populations across studies. Moreover, the scarcity of studies specifically focusing on the serotonergic system's role in time perception limits the available data for robust statistical pooling. A meta-analysis requires a sufficient number of high-quality, comparable studies to produce meaningful results, and the current state of the literature may not meet these criteria. Addressing these limitations would involve standardizing methodologies and increasing the volume of targeted research. Until such a foundation is established, a systematic review summarizing existing evidence might be more appropriate for synthesizing knowledge and identifying gaps in the field.
Comment 6. Since the current literature search was conducted in May 2023, and more than one and half year has passed. It would be better to update the literature search results to reflect the most recent update of this topic.
<< Response 6: We have addressed your concern and refreshed our search for the years 2023 and 2024 and found 7 articles (DOIs: 10.1038/s41380-024-02696-1, 10.1523/JNEUROSCI.0602-24.2024, 10.3389/fnbeh.2023.1289520, 10.1021/acsmedchemlett.3c00406, 10.1016/j.jpsychires.2023.08.005, 10.1016/j.cois.2023.101034, 10.1017/S0033291721005493).
Among them, only one article meets our inclusion criteria, and it was already included in our systematic review (Medvedeva et al., 2023). Thus, all the relevant existing studies are included. We have further specified in the 2.1. Literature Search Strategy section that we have refreshed the search: “The database search was conducted in May 2023 (and refreshed in December 2024) with no restrictions to the year of publication.”
Round 2
Reviewer 3 Report
Comments and Suggestions for Authors
Thanks for the revisions and one concern remains.
The Response 5 about meta-analysis can be incorporated into the limitation or future direction part.
Author Response
Comment 1: Thanks for the revisions and one concern remains.
The Response 5 about meta-analysis can be incorporated into the limitation or future direction part.
Response 1: We have included the discussion of a possibility to conduct a meta-analysis on the topic into the future directions part (before conclusions): “Another future direction on the level of knowledge systematization could be conducting a meta-analysis on the role of the serotonergic system in time perception. While it is theoretically possible, several challenges currently limit its feasibility. The field suffers from significant uncontrolled variability, such as differences in experimental designs, methodologies, and subject populations across studies. Moreover, the scarcity of studies specifically focusing on the serotonergic system's role in time perception limits the available data for robust statistical pooling. A meta-analysis requires a sufficient number of high-quality and comparable studies to produce meaningful results. Addressing these limitations would involve standardizing methodologies and increasing the volume of targeted research.”